# Small subpopulations of β-cells do not drive islet oscillatory [Ca$^{2+}$] dynamics via gap junction communication

**JaeAnn M. Dwulet**[1], **Jennifer K. Briggs**[1], **Richard K. P. Benninger**[1,2]*

**1** Department of Bioengineering, University of Colorado, Anschutz Medical campus, Aurora, Colorado, United States of America, **2** Barbara Davis center for childhood diabetes, University of Colorado, Anschutz Medical campus, Aurora, Colorado, United States of America

* richard.benninger@cuanschutz.edu

**Data Availability Statement:** The raw simulation data that supports the findings of this study are openly available at the BioStudies database under accession number S-BSST628 found at https://

## Abstract

The islets of Langerhans exist as multicellular networks that regulate blood glucose levels. The majority of cells in the islet are excitable, insulin-producing β-cells that are electrically coupled via gap junction channels. β-cells are known to display heterogeneous functionality. However, due to gap junction coupling, β-cells show coordinated [Ca$^{2+}$] oscillations when stimulated with glucose, and global quiescence when unstimulated. Small subpopulations of highly functional β-cells have been suggested to control [Ca$^{2+}$] dynamics across the islet. When these populations were targeted by optogenetic silencing or photoablation, [Ca$^{2+}$] dynamics across the islet were largely disrupted. In this study, we investigated the theoretical basis of these experiments and how small populations can disproportionality control islet [Ca$^{2+}$] dynamics. Using a multicellular islet model, we generated normal, skewed or bimodal distributions of β-cell heterogeneity. We examined how islet [Ca$^{2+}$] dynamics were disrupted when cells were targeted via hyperpolarization or populations were removed; to mimic optogenetic silencing or photoablation, respectively. Targeted cell populations were chosen based on characteristics linked to functional subpopulation, including metabolic rate of glucose oxidation or [Ca$^{2+}$] oscillation frequency. Islets were susceptible to marked suppression of [Ca$^{2+}$] when ~10% of cells with high metabolic activity were hyperpolarized; where hyperpolarizing cells with normal metabolic activity had little effect. However, when highly metabolic cells were removed from the model, [Ca$^{2+}$] oscillations remained. Similarly, when ~10% of cells with either the highest frequency or earliest elevations in [Ca$^{2+}$] were removed from the islet, the [Ca$^{2+}$] oscillation frequency remained largely unchanged. Overall, these results indicate small populations of β-cells with either increased metabolic activity or increased frequency are unable to disproportionately control islet-wide [Ca$^{2+}$] via gap junction coupling. Therefore, we need to reconsider the physiological basis for such small β-cell populations or the mechanism by which they may be acting to control normal islet function.

www.ebi.ac.uk/biostudies/studies/S-BSST628. The datasets are organized by figure.

**Funding:** This work was supported by Juvenile Diabetes Research Foundation (JDRF, https://www.jdrf.org/) Grant 5-CDA-2014-198-A-N (to RKPB); National Institute of Health (NIH, https://www.nih.gov/) grants R01 DK102950, R01 DK106412, R56 DK106412 (to RKPB); and NIH grant F31 DK126360 (to JMD). The funders had no role in the study design, data collection and analysis, decisions to publish, or preparation of the manuscript.

**Competing interests:** The authors have declared that no competing interests exist.

## Author summary

Many biological systems can be studied using network theory. How heterogeneous cell subpopulations come together to create complex multicellular behavior is of great value in understanding function and dysfunction in tissues. The pancreatic islet of Langerhans is a highly coupled structure that is important for maintaining blood glucose homeostasis. β-cell electrical activity is coordinated via gap junction communication. The function of the insulin-producing β-cell within the islet is disrupted in diabetes. As such, to understand the causes of islet dysfunction we need to understand how different cells within the islet contribute to its overall function via gap junction coupling. Using a computational model of β-cell electrophysiology, we investigated how small highly functional β-cell populations within the islet contribute to its function. We found that when small populations with greater functionality were introduced into the islet, they displayed signatures of this enhanced functionality. However, when these cells were removed, the islet, retained near-normal function. Thus, in a highly coupled system, such as an islet, the heterogeneity of cells allows small subpopulations to be dispensable, and thus their absence is unable to disrupt the larger cellular network. These findings can be applied to other electrical systems that have heterogeneous cell populations.

## Introduction

Many tissues exist as multicellular networks that have complex structures and functions. Multicellular networks are generally comprised of heterogenous cell populations, and heterogeneity in cellular function makes it difficult to understand the underlying network behavior. Studying the constituent cells individually is of value. However, understanding how heterogeneous cell populations come together to form a coherent structure with emergent properties is important to understand what leads to dysfunction in these networks [1]. The multicellular pancreatic islet lends itself to network theory with its distinct architecture, cellular heterogeneity, and cell-cell interactions.

The pancreatic islet is a micro-organ that helps maintain blood glucose homeostasis [2]. Death or dysfunction to insulin-secreting β-cells within the islet generally causes diabetes [3]. When blood glucose levels rise, glucose is transported into the β-cell and phosphorylated by glucokinase (GK), the rate limiting step of glycolysis [4–6]. Following glucose metabolism, the ratio of ATP/ADP increases, closing ATP sensitive K$^+$ channels (K$_{ATP}$). K$_{ATP}$ channel closure causes membrane depolarization, opening voltage gated Ca$^{2+}$ channels and elevating intra-cellular free-calcium ([Ca$^{2+}$]), which triggers insulin granule fusion and insulin release [7, 8]. Disruptions to this glucose stimulated insulin secretion pathway occur in diabetes [9–18]. β-cells are electrically coupled by connexin36 (Cx36) gap junctions which can transmit depolarizing currents across the islet that synchronize oscillations in [Ca$^{2+}$]. Under low glucose conditions, gap junctions transmit hyperpolarizing currents that suppress islet electrical activity [19–22]. Understanding the role cell-cell communication between β-cells plays can increase our understanding of dysfunction to islet dynamics during the pathogenesis of diabetes.

Despite their robust coordinated behavior within the intact islet, β-cells are functionally heterogeneous [23]. Individual β-cells show heterogeneity in expression of GK [24], glucose metabolism [23], differing levels of insulin production and secretion [25–28], and faster and irregular [Ca$^{2+}$] oscillations when compared to whole islet oscillations [29]. Various cell surface and protein markers have been used to identify subpopulations of β-cells with differences

in functionality and proliferative capacity [30–34]. Nevertheless, the importance of β-cell heterogeneity and how these subpopulations contribute to islet function is poorly understood.

While many studies of β-cell heterogeneity have been performed in dissociated cells, a few studies have investigated the role of heterogeneity in the intact islet [35]. In one study, following stimulation via the optogenetic cationic channel channelrhodopsin (ChR2), ~10% of β-cells were found to be highly excitable in that they are able to recruit $[Ca^{2+}]$ elevations in large regions of cells across the islet when stimulated at low glucose. These highly excitable cells had higher metabolic activity upon glucose elevation [36]. In another study, the optogenetic Cl⁻ pump halorhodopsin (eNpHr3) was used to silence β-cells. A population of ~1–10% "hub" β-cells was discovered that when hyperpolarized by eNpHr3 substantially disrupted coordinated $[Ca^{2+}]$ dynamics across the islet. These cells had increased GK expression [37]. In related studies, a small population of cells showed $[Ca^{2+}]$ oscillations that consistently preceded the rest of the islet and were suggested to be 'pacemaker cells' that drove islet $[Ca^{2+}]$ dynamics [38]. These cells that coincide with the initiation of the $[Ca^{2+}]$ wave were suggested to have higher intrinsic oscillation frequencies [36]. Theoretically, how small subpopulations of cells may be capable of driving elevations and oscillatory dynamics of $[Ca^{2+}]$ across the islet is not well established, and has been a significant topic of debate [39, 40].

In this study we explore the theoretical basis for whether small β-cell subpopulations can control multicellular islet $[Ca^{2+}]$ dynamics. Towards this, we utilize a computational model of the islet that we have previously validated against a wide-range of experimental data [36, 41–43]. This includes understanding how populations of inexcitable cells suppress islet function and the role for electrical coupling. We investigate whether small populations of highly metabolically active cells or cells with high frequency oscillations can respectively drive the elevations or dynamics of islet $[Ca^{2+}]$ oscillations. We systematically examined the effects of removal of specific cell populations within the context of broad normal distributions, skewed distributions or distinct bimodal distributions of heterogeneity. Our results indicate that small subpopulations of β-cells with increased metabolic activity or increased oscillation frequency are unable to drive islet $[Ca^{2+}]$ oscillations through gap junctional communication. Conversely, those cells with reduced metabolic activity or reduced oscillation frequency have a greater impact on islet $[Ca^{2+}]$ oscillations.

## Results

### How variation in metabolic activity impacts islet function

Experimental evidence indicates that within the intact islet there exists 10–20% variation in metabolic activity [44]. Previous modelling studies have represented beta cell heterogeneity as a unimodal normal distribution with 10–25% variation in GK activity and metabolic activity, which is sufficient to model the impact of electrical coupling and heterogeneity within the islet [36, 41–43]. However, recent experimental evidence has suggested that hub β-cells have elevated metabolic activity or GK expression, and this small population may disproportionately drive elevated $[Ca^{2+}]$ [36, 37].

We first asked whether identification of such a 'hub' subpopulations may arise as part of the natural variation within a unimodal normal distribution. We simulated an islet with a normal distribution in GK activity (Fig 1A), and targeted hyperpolarization to a population of cells based on their GK activity. Hyperpolarization as a result of current injection was used to mimic the optogenetic silencing that was performed in experimental studies [37]; where a key feature is an inhibitory current in the targeted cell that can suppress nearby cells via gap junction currents. Simulated islets had normal synchronized $[Ca^{2+}]$ oscillations (Fig 1B), comparable to previous studies [36, 41–43, 45, 46]. When hyperpolarization was targeted to a random

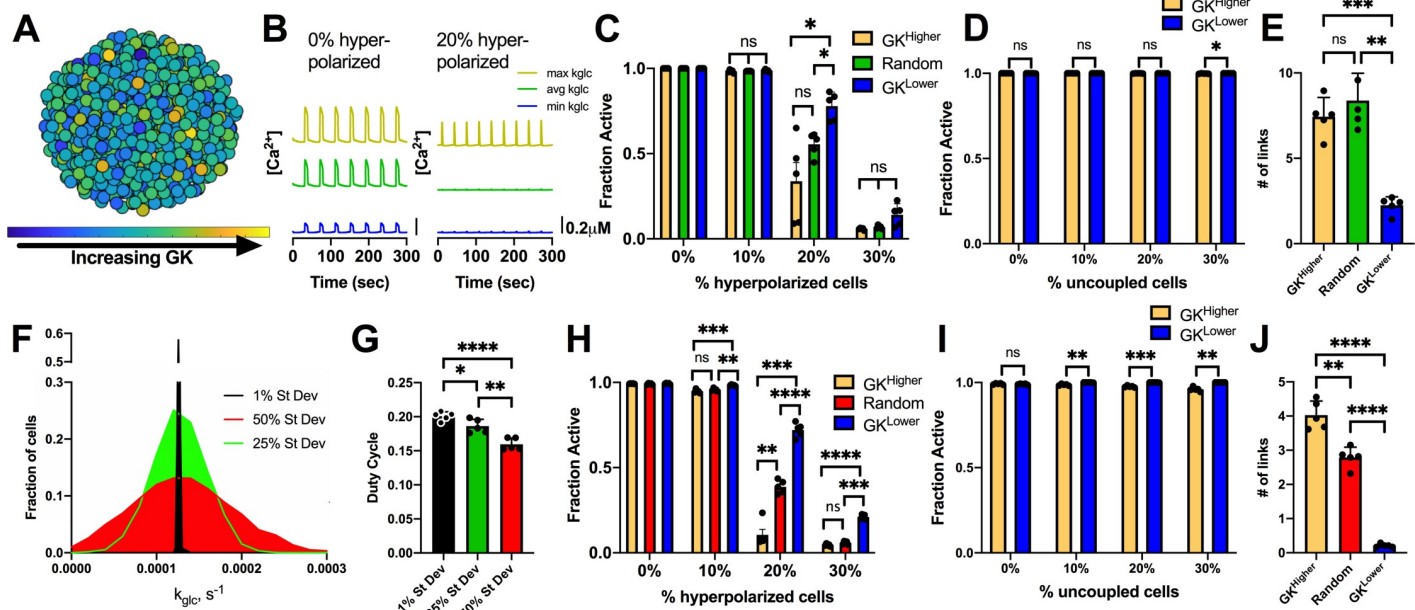

**Fig 1. Simulations predicting how variation in GK activity impact islet function.** A). Schematic of unimodal normal distribution of heterogeneous GK activity across simulated islet with 25% variation in GK rate ($k_{glc}$). B). Representative time courses of [Ca$^{2+}$] for 3 cells in simulated islet in A. Left is simulation with 0% hyperpolarized cells and right is simulation with a random 20% of cells hyperpolarized. Blue trace is cell with lowest GK rate ($k_{glc}$), Green is cell with the average GK rate, yellow is cell with the highest GK rate. C). Fraction of cells showing elevated [Ca$^{2+}$] activity (active cells) in simulated islets vs. the percentage of cells hyperpolarized in islet. Hyperpolarized cells are chosen based on their GK rate. D). Fraction of active cells in islet when cells are uncoupled from the rest of the cells in the simulation. E). # of links from network analysis of [Ca$^{2+}$] activity with GK 25% variation. F). Histogram showing average frequency of cells at varying GK rate ($k_{glc}$) for simulations that have different standard deviation in GK activity. G). Average duty cycle of cells from simulations with different standard deviation in GK activity. H). As in C. for simulations with standard deviation in GK activity at 50% of the mean. I). As in D. for simulations with standard deviation in GK activity at 50% of the mean. J). As in E but for simulations with GK 50% variation. Error bars are mean ± s.e.m. Repeated measures one-way ANOVA with Tukey post-hoc analysis was performed for simulations in C and G, Student's paired t-test was performed for D and H, and one-way ANOVA was performed for F to test for significance. Significance values: ns indicates not significant (p>.05), * indicates significant difference (p < .05), ** indicates significant difference (p < .01), *** indicates significant difference (p < .001), **** indicates significant difference (p < .0001). Data representative of 5 simulations with differing random number seeds.

set of cells across the islet, near-normal [Ca$^{2+}$] activity was maintained until greater than 20% of cells within the islet were targeted (Fig 1B and 1C). Above this level, the islet lacked significant [Ca$^{2+}$] elevations (Fig 1C), consistent with prior measurements [41, 43]. When hyperpolarization was targeted specifically to cells with either higher GK (GK$^{Higher}$) or lower GK (GK$^{Lower}$), similar changes in [Ca$^{2+}$] activity were observed as with targeting a random subset of cells: the islet retained near-normal [Ca$^{2+}$] activity until greater than 20% of these GK$^{Higher}$ or GK$^{Lower}$ cells were targeted (Fig 1C). Nevertheless when 20% of cells were hyperpolarized, targeting GK$^{Higher}$ cells did result in silencing of significantly more of the islet compared to GK$^{Lower}$ cells. Within the simulated islet, we also decoupled and removed the same GK$^{Higher}$ or GK$^{Lower}$ populations. In this case, the remaining islet showed normal elevations in [Ca$^{2+}$], with little to no difference between removing GK$^{Higher}$ or GK$^{Lower}$ cells (Fig 1D). We performed network analysis [37] to test whether cells with higher GK activity (GK$^{Higher}$) or lower GK activity (GK$^{Lower}$) show differing connectivity. GK$^{Higher}$ cells showed an increased proportion of links compared to GK$^{Lower}$ cells (Fig 1E). Therefore, our current simulated islet does not accurately describe the behavior of small highly functional subpopulations identified from previous experiments.

Given uncertainty in the exact level of heterogeneity within the islet, we next tested whether changes to the variability in GK could lead to differences in [Ca$^{2+}$] upon targeting cells with higher GK (GK$^{Higher}$) or lower GK (GK$^{Lower}$) cells. We simulated islets with decreased

variation in GK activity (1% variation) or increased variation in GK activity (50% variation) and compared [Ca$^{2+}$] with our previous simulations of 25% variation (Figs 1F and S1A). The duty cycle of the simulated islets slightly decreased as the GK variation increased (Fig 1G), but [Ca$^{2+}$] oscillations remained across the islet that closely matched previous studies. Under 50% variation in GK, when hyperpolarization was targeted to a random set of cells across the islet, the islet retained near-normal [Ca$^{2+}$] activity until greater than 20% of the islet was targeted, as before. In contrast, when hyperpolarization was targeted specifically to cells with higher GK (GK$^{Higher}$), [Ca$^{2+}$] was largely abolished for greater than 10% of cells being targeted (Fig 1H). However, when hyperpolarization was targeted to lower GK (GK$^{Lower}$) cells, [Ca$^{2+}$] was largely unchanged until 30% of cells were targeted (Fig 1H). As such, upon hyperpolarizing 20% of cells, a substantial difference in [Ca$^{2+}$] resulted from targeting GK$^{Higher}$ or GK$^{Lower}$ cells. Nevertheless, when these higher GK or lower GK cells were decoupled and removed from the islet, the impact on [Ca$^{2+}$] elevations was very minor. A minor 2–4% decrease in [Ca$^{2+}$] occurred when removing >10% GK$^{Higher}$ cells, with no impact when removing GK$^{Lower}$ cells (Fig 1I). Following network analysis with increased variation in GK activity, GK$^{Higher}$ cells showed an increased proportion of links compared to both random cells within the islet and compared to GK$^{Lower}$ cells (Fig 1J). Finally, we investigated whether stochastic noise could impact these results. When noise was incorporated, there was little difference in the threshold number of cells needed to be hyperpolarized to suppress islet activity under either 25% or 50% variation in GK activity (S2A and S2B Fig).

We also tested whether changing other properties of cells with higher GK or lower GK would impact the suppression of [Ca$^{2+}$]. When GK activity correlated with gap junction conductance such that higher GK cells also had increased gap junction conductance (GK$^{Higher}$/g$_{Coup}$$^{Higher}$), little impact was observed (S3A–S3C Fig): more than 20% of GK$^{Higher}$/g$_{Coup}$$^{Higher}$ cells were still needed to be hyperpolarized to fully silence the islet. However, when hyperpolarizing 20% of GK$^{Lower}$/g$_{Coup}$$^{Lower}$ cells, [Ca$^{2+}$] was largely unchanged. Little difference was observed when GK activity negatively correlated with K$_{ATP}$ conductance, such that higher GK cells also had reduced K$_{ATP}$ conductance (S3D–S3F Fig).

Thus, hyperpolarizing a small subpopulation of metabolically active cells can disproportionately suppress islet [Ca$^{2+}$], particularly when heterogeneity is very broad or GK activity was correlated with other beneficial factors. However, when these same cells are removed or absent from the islet, the impact on [Ca$^{2+}$] is minimal under the model assumptions set here.

## Impact of alternative distributions of functional β-cell subpopulations

We next examined how imposing a unimodal skewed or bimodal distribution in GK activity would impact targeting hyperpolarization to a small population of metabolically active cells. We simulated an islet with a unimodal skewed distribution, which resulted in a population of highly metabolic cells that comprised 10% of the islet and had ~3 times the GK activity (GK$^{High}$) (Fig 2A). The mean GK activity was equivalent to previous studies, such that the remainder of the islet had slightly reduced GK activity (GK$^{Low}$) (Fig 2B). Gap junction coupling conductance of all cells remained unchanged (S1B Fig). Under this skewed distribution, the islet displayed regular [Ca$^{2+}$] oscillations at high glucose, but with slightly lower duty cycle compared to simulations using a unimodal normal distribution (Fig 2C). We tested the effect of targeting hyperpolarization to either the GK$^{High}$ or GK$^{Low}$ cell populations. When all GK$^{High}$ cells (10%) were hyperpolarized, [Ca$^{2+}$] was fully suppressed across the islet. Conversely, when GK$^{Low}$ cells (10%) were hyperpolarized, [Ca$^{2+}$] showed reduced suppression (~40% activity) (Fig 2D). When a greater proportion of GK$^{Low}$ cells (20%) were hyperpolarized, [Ca$^{2+}$] was suppressed, as with a unimodal normal distribution. Following network analysis, GK$^{High}$ cells showed an increased proportion of links, indicating increased connectivity

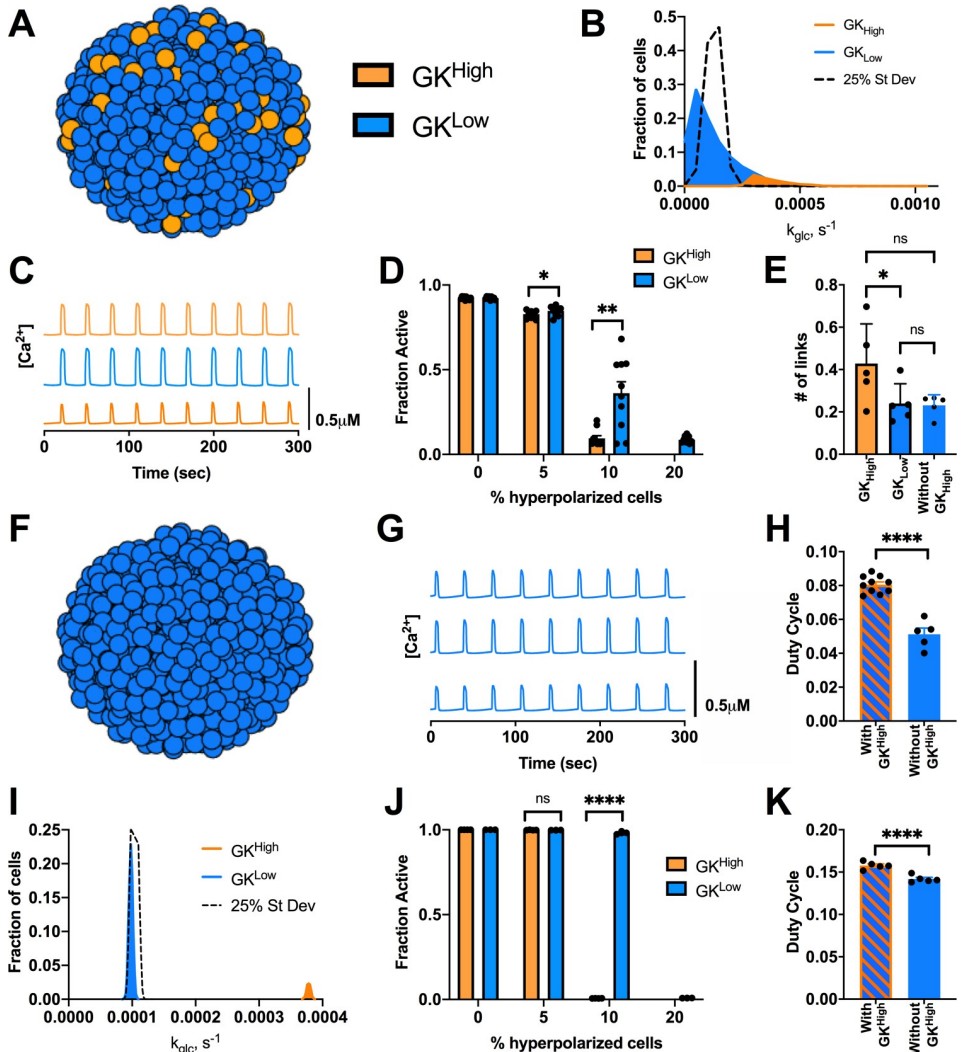

**Fig 2. Alternative distributions in GK activity predicts small highly functional cells are dispensable for islet [Ca²⁺] dynamics.** A). Schematic of altered distributions of GK activity across simulated islet. B). Histogram showing average frequency of cells at varying GK rate ($k_{glc}$) for skewed distribution compared with normal distribution (25% St Dev). C). Representative time courses of [Ca²⁺] for 3 cells in simulated skewed islet in A. Blue traces are cells from GK$^{Low}$ population and orange traces are cells from GK$^{High}$ population. D). Fraction of cells showing elevated [Ca²⁺] activity (active cells) in skewed simulations vs. the percentage of cells hyperpolarized in islet. Hyperpolarized cells are chosen either from GK$^{High}$ (orange bars) or GK$^{Low}$ (blue bars) population. E). # of links from network analysis of [Ca²⁺] activity for simulations with skewed distribution of GK F). Schematic of simulation where only GK$^{Low}$ cells are present and no GK$^{High}$ cells are included. G). Representative time courses of [Ca²⁺] for 3 cells in simulated islet in F. H). Average duty cycle of cells from simulations of a skewed distribution model as in A (With GK$^{High}$) and from simulations as in F (Without GK$^{High}$). I). As in B. for bimodal distribution in GK activity where the populations only have 2.5% variation in GK activity. J). As in D. for bimodal distribution in GK activity. K). As in H. but for simulations with no GK$^{High}$ from a bimodal distribution. Error bars are mean ± s.e.m. Student's paired t-test was performed to test for significance. Significance values: ns indicates not significant (p>.05), * indicates significant difference (p < .05), ** indicates significant difference (p < .01), *** indicates significant difference (p < .001), **** indicates significant difference (p < .0001). Data representative of 4–9 simulations with differing random number seeds.

as measured in hub cells (Fig 2E). Again, we tested the effect of noise under this new distribution and only slight differences were observed (S2C Fig). These results show good agreement between prior experiments where a small population of cells with high GK activity was able to greatly reduce the [Ca²⁺] response under hyperpolarization.

We next tested whether the cells from the highly metabolic population (GK$^{High}$) are important to support islet function, by simulating an islet with only cells from the lower GK population (GK$^{Low}$) (Fig 2F). With no GK$^{High}$ cells present, the islet [Ca$^{2+}$] activity still displayed oscillations (Fig 2G), but duty cycle decreased by ~40% (Fig 2H). The number of links for the GK$^{Low}$ remained unchanged even when GK$^{High}$ cells were removed (Fig 2E). These results indicate that [Ca$^{2+}$] is maintained across the islet in the absence of a small population (10%) of highly metabolic cells, suggesting these cells are not required to drive elevated [Ca$^{2+}$].

Next, we tested a bimodal distribution where the two populations are substantially different. The GK activity for GK$^{High}$ cells was still ~3 times the overall mean GK activity, with the overall mean GK activity unchanged. However, each population had a distinct normal distribution with 2.5% variation (Fig 2I). Gap junction coupling remained unchanged (S1D Fig). Under this bimodal distribution when all GK$^{High}$ cells (10%) were hyperpolarized, [Ca$^{2+}$] was fully suppressed across the islet. Conversely, when GK$^{Low}$ cells (10%) were hyperpolarized, [Ca$^{2+}$] remained largely unchanged (Fig 2J). However, when a greater proportion of GK$^{Low}$ cells (20%) were hyperpolarized, [Ca$^{2+}$] was suppressed, as with a unimodal normal distribution under 50% variation. These results show very good agreement between prior experiments, where very different [Ca$^{2+}$] response was observed when hyperpolarizing cells with higher GK and cells with lower GK. However, when GK$^{High}$ cells were removed from the bimodal distribution, the islet retained near-normal [Ca$^{2+}$] activity, with a minor (~10%) drop in duty cycle (Fig 2K). As such, the simulated islet was capable of maintaining normal elevated [Ca$^{2+}$] in the absence of a small (~10%) highly metabolic subpopulation. Thus, despite showing substantial differences in islet activity when hyperpolarized, a small metabolically active subpopulation is not required to maintain elevations in oscillatory [Ca$^{2+}$] across the islet.

## How variations in gap junction coupling impact functional β-cell subpopulations

Metabolically active subpopulations of cells that disproportionately control the islet have increased connectivity [37]. This has been suggested to result from increased gap junction coupling. We next examined how changes in gap junction electrical coupling affect how targeting hyperpolarization to specific cell populations impacts islet [Ca$^{2+}$]. We simulated the islet with the same skewed distribution in GK activity as in Fig 2B, but correlated gap junction coupling conductance (g$_{Coup}$) with GK activity (k$_{glc}$) across the islet (Figs 3A and S1C). As such, more metabolically active GK$^{High}$ cells had ~2 times higher gap junction conductance than that of the population of cells with lower metabolic activity (GK$^{Low}$ cells). GK$^{High}$ cells, which had higher coupling, did not show significantly different suppression of islet Ca$^{2+}$ following hyperpolarization compared to GK$^{Low}$ cells that had reduced coupling (Fig 3B); unlike the significant differences when coupling was uniform. Thus, increased coupling does not allow GK$^{High}$ cells to impact the islet to a greater degree upon hyperpolarization (Fig 3C). Next, we correlated coupling with GK activity under a bimodal distribution in GK activity as in Fig 2I (Figs 3D and S1E). Under this model, when 20% of the highly metabolic GK$^{High}$ cells were targeted with hyperpolarization, the islet retained [Ca$^{2+}$] elevations (~25%) (Fig 3E). When GK$^{Low}$ cells with less metabolic activity were targeted with hyperpolarization, the islet also showed less [Ca$^{2+}$] activity compared to previous simulations. As such, the difference in suppression of [Ca$^{2+}$] upon targeting hyperpolarization to either population is reduced when highly metabolic cells having elevated electrical coupling (Fig 3F). Thus, increasing gap junction coupling does not enhance the ability of metabolically active cells to maintain oscillatory islet [Ca$^{2+}$] elevations when compared with lower metabolic cells. Unexpectedly, increasing coupling in

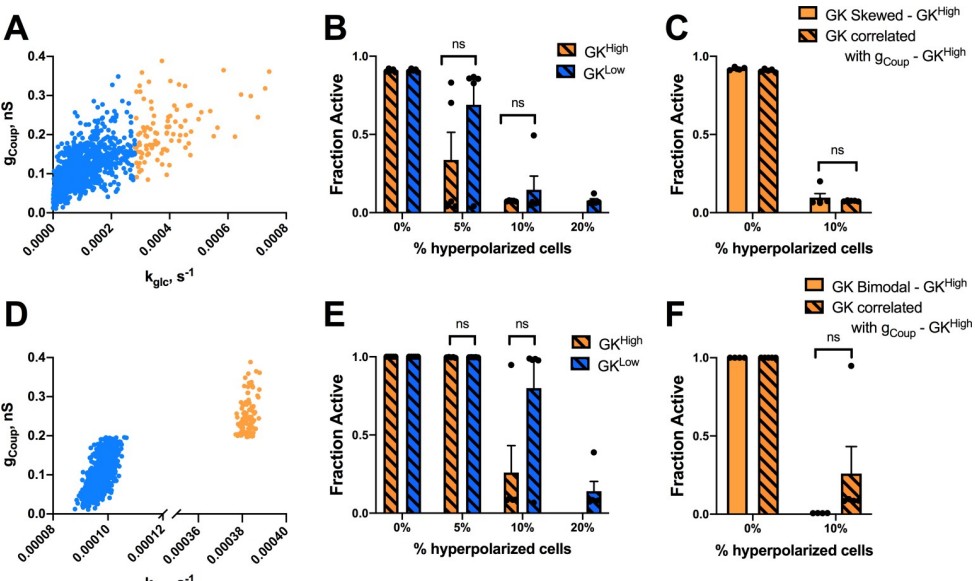

**Fig 3. Simulations predicting how changes in coupling impact highly metabolic populations.** A). Scatterplot of $g_{Coup}$ vs. $k_{glc}$ for each cell from a representative simulation where $g_{Coup}$ is correlated with $k_{glc}$ under a skewed distribution in GK activity. B). Fraction of cells showing elevated $[Ca^{2+}]$ activity (active cells) vs. the percentage of cells hyperpolarized in islet from skewed simulations in $k_{glc}$ with correlated $g_{Coup}$ and $k_{glc}$ as in A. Hyperpolarized cells are chosen either from $GK^{High}$ (orange bars) or $GK^{Low}$ (blue bars) population. C). As in B. but comparing hyperpolarization in $GK^{High}$ cells in the presence and absence of correlations in $g_{Coup}$. D). as in A but from a simulation where $g_{Coup}$ and $k_{glc}$ are correlated under a bimodal distribution in GK activity. E). As in B. but for simulations where $g_{Coup}$ and $k_{glc}$ are correlated under a bimodal distribution in GK activity. F). As in C. but comparing hyperpolarization in $GK^{High}$ cells in the presence and absence of correlations in $g_{Coup}$ under a bimodal distribution in GK activity. Error bars are mean ± s.e.m. Student's paired t-test was performed for B and E and a Welches t-test for unequal variances was used for C and F to test for significance. Significance values: ns indicates not significant (p>.05), * indicates significant difference (p < .05), ** indicates significant difference (p < .01), *** indicates significant difference (p < .001), **** indicates significant difference (p < .0001). Data representative of 4 simulations with differing random number seeds.

metabolically active cells and decreasing coupling in cells with decreased metabolic activity causes the two populations to act more similarly in their control over islet $[Ca^{2+}]$.

Given this dependence on gap junction coupling, we examined whether decreases in coupling impacted how metabolically active cells controlled islet function. We performed similar simulations as in Figs 1 and 2 for an islet with reduced average gap junction conductance of 50%. In this context, hyperpolarizing highly metabolic populations ($GK^{Higher}$ or $GK^{High}$) or cells with reduced metabolic activity ($GK^{Lower}$ or $GK^{Low}$) reduced islet $[Ca^{2+}]$ to a lesser degree than when gap junction conductance was higher (S4 Fig). This applied to simulated islets with either a unimodal normal distribution in GK activity (S4A and S4B Fig) or a bimodal distribution of GK activity (S4C and S4D Fig). In each case, a similar difference in islet $[Ca^{2+}]$ resulted from hyperpolarizing highly metabolic or low metabolic cells, albeit with greater numbers of cells needing targeting to suppress $[Ca^{2+}]$. Thus, decreasing gap junction coupling does not enhance the ability of small populations of metabolic active cells to maintain islet $[Ca^{2+}]$.

## Cells with $[Ca^{2+}]$ oscillations that precede the rest of the islet do not drive islet $[Ca^{2+}]$ oscillations

Another subpopulation of β-cells that has been associated with islet function are those cells that show $[Ca^{2+}]$ oscillations that precede oscillations across the rest of the islet [36, 38, 47].

These cells have been suggested to have higher intrinsic oscillation frequency [36, 38], which may lend themselves to act as rhythmic pacemakers to drive [Ca²⁺] oscillations across the islet. We next investigated whether a small subpopulation of these cells is able to drive islet [Ca²⁺] oscillatory dynamics. We simulated an islet with a unimodal normal distribution of heterogeneity, as in Fig 1, and identified cells with [Ca²⁺] oscillations that preceded the rest of the islet (early phase) or cells with [Ca²⁺] oscillations that are delayed with respect to the rest of the islet (late phase) (Fig 4A and 4B). Cells that preceded the rest of the islet (early phase cells) were temporally separated to a greater degree with respect to the rest of the islet compared to cells that were delayed (late phase cells) (Fig 4C). The top 1% and 10% of early phase cells (earlier [Ca²⁺] oscillations) in the islet had higher intrinsic oscillation frequency–the oscillation frequency if the cell is simulated in isolation–and lower GK activity compared to the rest of the islet (Fig 4D and 4E). This is consistent with prior experimental measurements that demonstrated lower metabolic activity in cells that show earlier [Ca²⁺] oscillations [36]. Conversely, the top 1% and 10% of late phase cells (delayed [Ca²⁺] oscillations) had lower intrinsic oscillation frequency and high GK activity (Fig 4D and 4E).

To determine the role these cells may play in islet function, we re-simulated the islet with populations of early phase and late phase cells removed from the islet. When populations (1%, 10%, 30%) of early or late phase cells were removed, the elevation of [Ca²⁺] was unchanged (S5A Fig). Similarly, the frequency of the islet did not differ significantly from control islets when up to 10% of early or late phase cells were removed (Fig 4F and 4G). Early or late phase cells usually exist within a compact region, rather than being distributed randomly across the islet. Removing random cells within a similar sized region impacts frequency of the remaining islet to a lesser degree than removing randomly positioned cells across the islet (S6 Fig). Removal of up to 10% of early phase or late phase cells also showed no change in frequency compared to removal of random cells within a similar sized region (Fig 4F and 4G). When 30% of early phase cells (earlier [Ca²⁺] oscillations) were removed from the islet, frequency decreased slightly, by ~2% (Fig 4G). This minor decrease in frequency was equivalent to the average frequency of the remaining cells in the islet, indicating no disproportionate effect of the early phase cells on oscillation frequency (Fig 4H). In contrast, when 30% of late phase cells (delayed [Ca²⁺] oscillations) were removed, the islet frequency increased, by ~8% (Fig 4G). This increase in frequency upon removing the late phase cells was significantly greater than the average frequency of the remaining cells in the islet, indicating a disproportionate effect of late phase (delayed) cells on oscillation frequency (Fig 4I). When these manipulations were performed in the presence of reduced (50%) gap junction conductance, the changes in frequency were exacerbated: no change in frequency when removing early phase cells and a greater increase in frequency (~15%) when removing late phase (delayed) cells (S7 Fig). Finally, early phase cells did not show a significant difference in the number of links compared with late phase cells in the islet (Fig 4J). Thus, early phase cells that show earlier [Ca²⁺] oscillations do not drive the [Ca²⁺] oscillation frequency of the islet, when considering a unimodal normal distribution of cell heterogeneity. However, unexpectedly, late phase cells that show delayed [Ca²⁺] oscillations appear to drive a slower [Ca²⁺] oscillation frequency; but only in proportions of at least 30% of the islet.

Early phase and late phase cells that show different timings in their [Ca²⁺] oscillations on average have higher or lower intrinsic [Ca²⁺] oscillation frequency respectively. However, other factors such as gap junction coupling or position within the cluster may also determine their relative timing. We next examined the role of cells that intrinsically have the highest or lowest [Ca²⁺] oscillation frequency (Fig 5A–5C). The top 1% or 10% of cells with highest or lowest intrinsic oscillation frequency, showed a frequency substantially different than the islet average (Fig 5D). On average, cells with a higher intrinsic oscillation frequency showed earlier

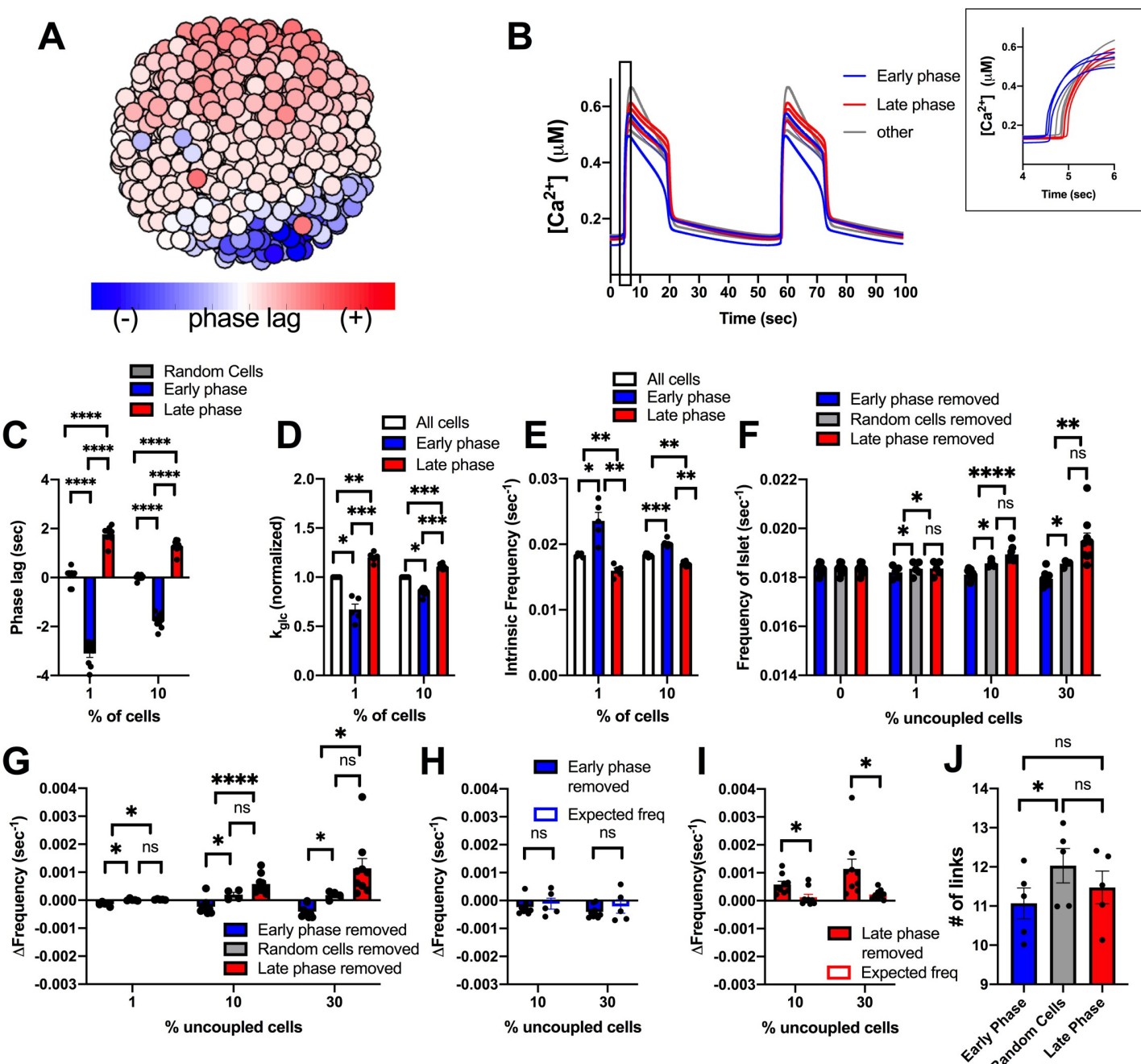

**Fig 4. Simulations predicting how small populations of cells contribute to islet frequency.** A). Schematic of phase lag across simulated islet with 25% variation in GK activity. B). Representative time courses of [Ca²⁺] for 9 cells in simulated islet at 60pS coupling conductance to determine phase lag of cells in A. Blue traces are early phase cells (negative phase lag), Grey is non early or late phase cells, red is late phase cells (positive phase lag). Inset: Close up of rise of [Ca2+] oscillation showing phase lags. C). Phase lag from islet average of top 1% or 10% of early phase, late phase cells, or random cells. D). Average $k_{glc}$ from all cells, early phase cells or late phase cells across simulated islet (normalized to average $k_{glc}$). E). Average intrinsic oscillation frequencies of all cells, top 1% and 10% of early phase cells, or top 1% or 10% of late phase cells when re-simulated in the absence of gap junction coupling (0pS). F). Average frequency of islet when indicated populations of cells are removed from the simulated islet. G). Change in frequency of islet with indicated populations removed with respect to control islet with all cells present. H). Change in frequency when early phase cells are removed compared to average oscillation frequency of remaining cells that indicates the expected oscillation frequency. I). Same as H. but for simulations where late phase cells are removed. J). # of links from network analysis for early phase, late phase, and random cells in simulations. Error bars are mean ± s. e.m. Repeated measures one-way ANOVA with Tukey post-hoc analysis was performed for simulations in C-G (if there were any missing values a mixed effects model was used), Student's paired t-test was performed for H and I to test for significance. Significance values: ns indicates not significant (p>.05), * indicates significant difference (p < .05), ** indicates significant difference (p < .01), *** indicates significant difference (p < .001), **** indicates significant difference (p < .0001). Data representative of 4–9 simulations with differing random number seeds. Random regions were removed for 10% and 30% simulations, but random removal of cells was used for 1% simulations.

[Ca²⁺] oscillations compared with the average of the islet (Fig 5E) and had lower metabolic activity (Fig 5F). In contrast, cells with the lowest frequency showed delayed [Ca²⁺] oscillations compared with the average of the islet and had higher metabolic activity (Fig 5E and 5F). This is consistent with previous experimental measurements that demonstrated a negative correlation between oscillation frequency and metabolic activity [36]. We do note that a small (~0.5%) of cells with low metabolic activity lacked [Ca²⁺] elevations and were excluded from frequency measurements.

When greater than 10% or 30% of high frequency cells were removed from the islet, the frequency of the islet decreased, whereas when 10% or 30% of lower frequency cells were removed from the islet, the frequency of the islet increased (Fig 5G and 5H). However, in each case the change in frequency upon removing high or low frequency cells was not significantly greater than the change when considering the average frequency of the remaining cells (Fig 5I and 5J). In fact, the decrease in frequency upon removing high frequency cells was significantly less than that considering the frequency of remaining cells (Fig 5I). In each case, the elevation of [Ca²⁺] was unchanged (S5B Fig). These results again suggest that small numbers of cells with faster oscillation frequency do not disproportionately affect islet [Ca²⁺] oscillations.

## A bimodal distribution in frequency lessens the effect of late phase cells

Earlier we considered a bimodal distribution in metabolic activity that better described experimental data (Fig 2) [37]. We next investigated whether late phase and early phase cells may influence the islet to a greater degree when described by a bimodal distribution. From the unimodal normal distribution, we previously modelled (Fig 4), we generated a population of cells that incorporated the average properties of early phase cells that showed earlier oscillations in [Ca²⁺] (see methods). This population (10%), which showed a faster oscillation frequency (Figs 6A and S8A) was combined with a population of cells that were similar to the average properties of an islet. The resultant simulated islet showed cells with earlier and delayed [Ca²⁺] oscillations, as before (Fig 6B and 6C), albeit with a slight reduction in the time between the early and delayed oscillations (Fig 6D). On average, the early phase cells that showed earlier [Ca²⁺] oscillations had higher intrinsic oscillation frequencies (Fig 6E) and lower metabolic activity (Fig 6F), as before. However, the difference between early and late phase cells was not a large as with the unimodal normal distribution. When early phase cells or late phase cells were removed from the islet, the frequency was not significantly different than when random cells were removed (Fig 6G). However, when 10% of early phase cells were removed, the change in frequency was significantly different, albeit small, compared to the expected frequency of the remaining cells in the distribution (Fig 6H). On the other hand, the removal of late phase cells was not significantly different than the expected frequency of the remaining cells (Fig 6I).

We further examined how the islet behaved when the high frequency population of cells were removed. These high frequency cells showed only slightly earlier [Ca²⁺] oscillations compared to the rest of the islet on average (S8B Fig) but did show lower metabolic activity (S8C Fig). Upon removal of these high frequency cells, the islet showed significantly slower oscillations (S8D Fig), that were slower than expected given the average frequency of the remaining cells (S8D and S8E Fig). However, the change in frequency was still low (~2%). When these high frequency cells were positioned with the same spatial distribution as early phase cells, the change in frequency upon their removal was significantly greater but was still relatively small and similar to the change seen when high frequency cells were removed from the unimodal normal distribution model (~5%) (S9 Fig). In conclusion, within a bimodal distribution, a small population of cells with higher frequencies has only a minor impact on the frequency of the islet.

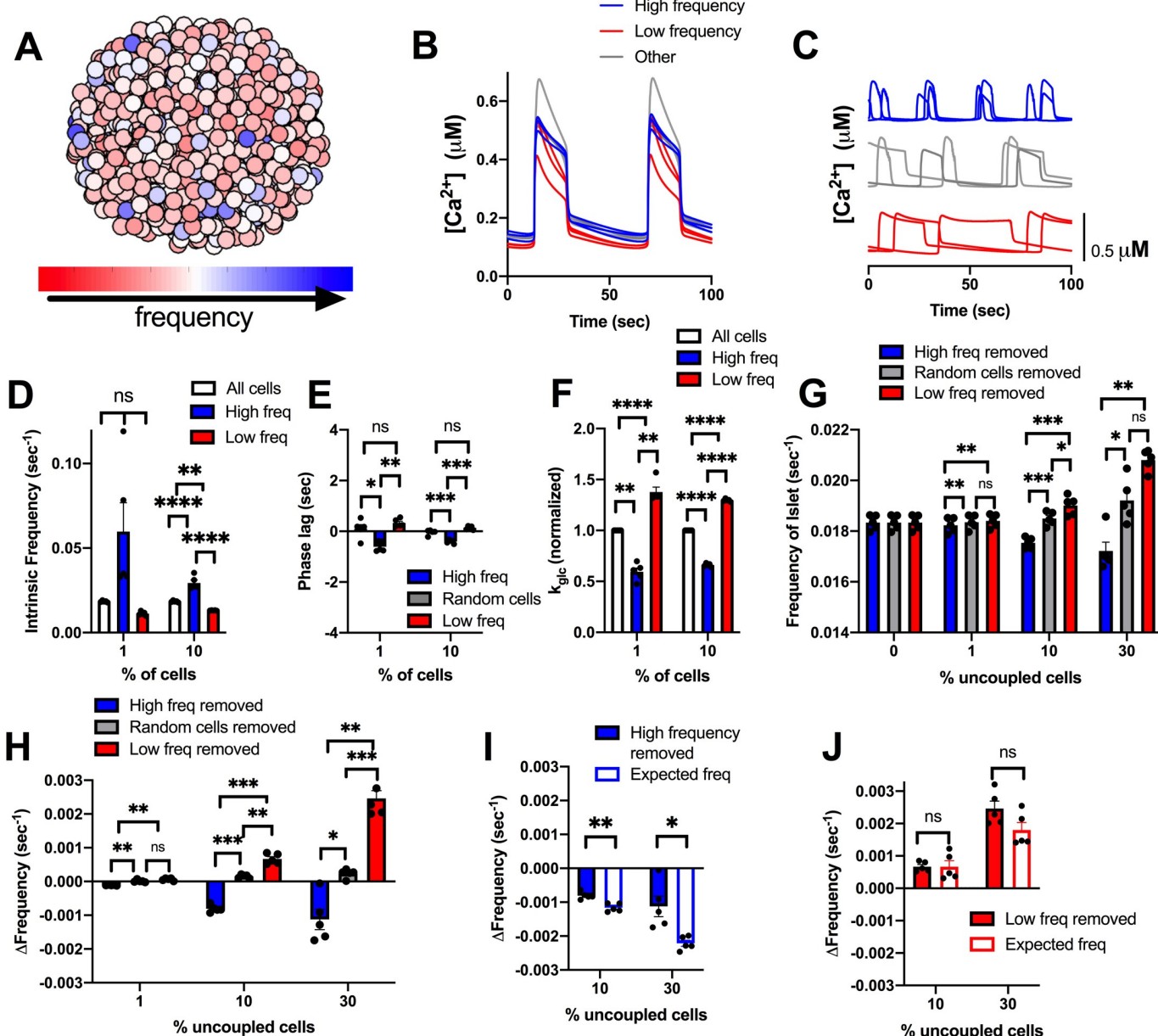

**Fig 5. Simulations predicting how intrinsic frequency of cells contributes to islet frequency.** A). Schematic of frequency across simulated islet with 25% variation in GK activity. B). Representative time courses of [Ca²⁺] for 9 cells in simulated islet in A in a simulation with full (120pS) coupling conductance. Blue traces are high frequency cells, Grey are cells with frequency near average frequency, red traces are low frequency cells. C). Same cells as in B but showing [Ca²⁺] time courses from an uncoupled simulation (0pS coupling conductance). D). Average intrinsic oscillation frequencies of all cells, top 1% or 10% of high frequency cells, or low frequency cells when re-simulated in the absence of gap junction coupling. E). Phase lag from islet average of top 1% or 10% of low frequency, high frequency, or random cells. F). Average $k_{glc}$ from all cells, high frequency cells, or low frequency cells across simulated islet (normalized to average $k_{glc}$). G). Average frequency of islet when indicated populations of cells are removed from the simulated islet. H). Change in frequency of islet with indicated populations removed with respect to control islet with all cells present. I). Change in frequency when high frequency cells are removed compared to average oscillation frequency of remaining cells that indicates the expected oscillation frequency. J). Same as I. but for simulations where low frequency cells are removed. Error bars are mean ± s.e.m. Repeated measures one-way ANOVA with Tukey post-hoc analysis was performed for simulations in D-H and a Student's paired t-test was performed for I and J to test for significance. Significance values: ns indicates not significant (p>.05), * indicates significant difference (p < .05), ** indicates significant difference (p < .01), *** indicates significant difference (p < .001), **** indicates significant difference (p < .0001). Data representative of 5 simulations with differing random number seeds. Random removal of cells across the islet was used for all simulations where random cells removed is indicated.

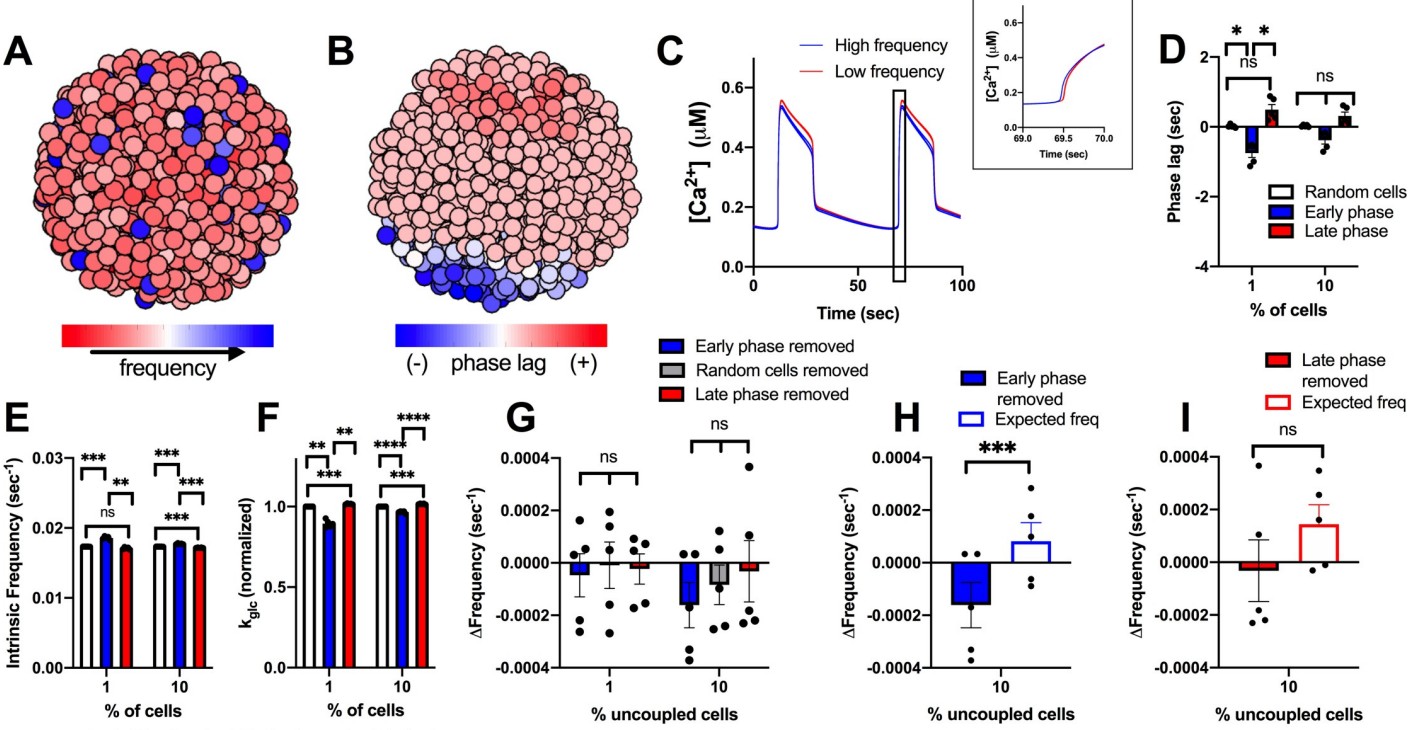

**Fig 6. Simulations predicting how cells from a bimodal distribution characterized by early phase cells contribute to islet frequency.** A). Schematic of frequency across simulated islet with a bimodal distribution in GK activity. B). Schematic of phase lag across simulated islet with a bimodal distribution in GK activity. C). Representative time courses of [Ca²⁺] for 6 cells in simulated islet in A (and B) in a simulation with full (120pS) coupling conductance. Blue traces are high frequency cells, red traces are low frequency cells. Inset: Close up of rise of [Ca2+] oscillation showing phase lags. D). Phase lag from islet average of top 1% or 10% of early phase, late phase cells, or random cells. E). Average intrinsic oscillation frequencies of all cells and 1% or 10% of early phase cells, or 1% or 10% of late phase cells when re-simulated in the absence of gap junction coupling (0pS). F). Average $k_{glc}$ from all cells and top 1% or 10% of early phase cells or late phase cells across simulated islet (normalized to average $k_{glc}$). G). Change in frequency of islet with indicated populations removed with respect to control islet with all cells present. H). Change in frequency when early phase cells are removed compared to average oscillation frequency of remaining cells that indicates the expected oscillation frequency. I). Same as H. but for simulations where late phase cells are removed. Repeated measures one-way ANOVA with Tukey post-hoc analysis was performed for simulations in D-G and a Student's paired t-test was performed for H and I to test for significance. Error bars are mean ± s.e.m. Significance values: ns indicates not significant (p>.05), * indicates significant difference (p < .05), ** indicates significant difference (p < .01), *** indicates significant difference (p < .001), **** indicates significant difference (p < .0001). Data representative of 5 simulations with differing random number seeds. Random removal of cells across the islet was used where random cells removed is indicated.

## Limited excitatory gap junction current can explain lack of action of small subpopulations

To understand the basis by which cells with differing metabolic activity and oscillatory frequency interact, we examined the gap junction currents for cell populations within the islet (Fig 7A). As expected, the total membrane current was highest in magnitude during the upstroke and downstroke of the [Ca²⁺] oscillation, and low in magnitude during the active and silent phase (Fig 7B–7D). Conversely, the gap junction current was highest during the active and silent phase of the [Ca²⁺] oscillation but was minimal during the upstroke and downstroke of [Ca²⁺] (Fig 7B, 7C and 7E). Thus, there is less communication between cells during the upstroke and downstroke of [Ca²⁺] oscillations compared to the stable active and silent phases.

The total membrane current did not differ significantly between cells with high or low metabolic activity (Fig 7D). However, there was a substantial difference in gap junction current between cells with high or low metabolic activity (Fig 7E). Cells with high metabolic activity

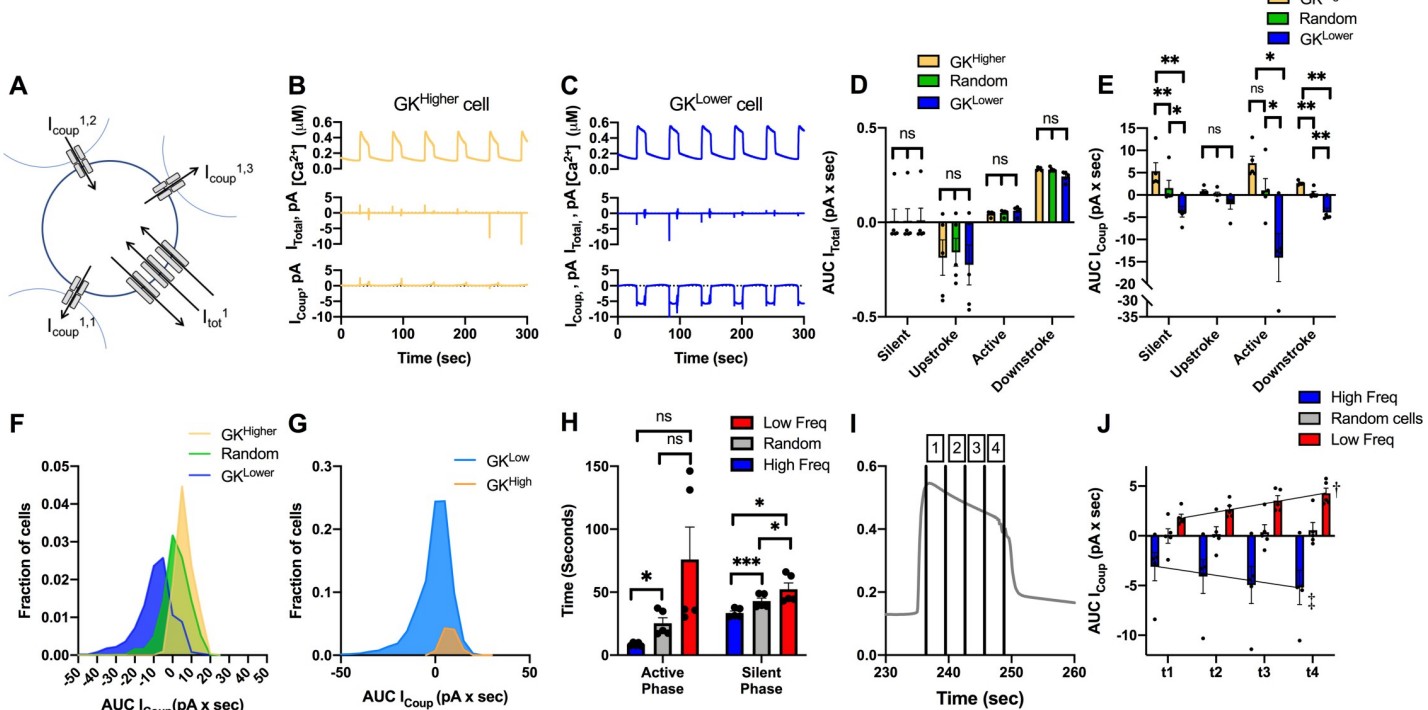

**Fig 7. Gap junction current in cells with high/low metabolic activity or oscillation frequency.** A). Schematic of cell within the simulated islet, showing 3 gap junction currents that contribute to the total gap junction current, together with the total membrane current. B). Time course of [Ca²⁺] from a cell, together with the total membrane current and total gap junction current for a representative cell with higher metabolic activity ($k_{glc}$). C). As in B for a representative cell with lower metabolic activity. D). Total membrane current, as expressed by an area under the curve (AUC), for each phase of the [Ca²⁺] oscillation averaged over the 10% of cells with highest or lowest $k_{glc}$ or a random 10% of cells. E). As in D for total gap junction current. F). Distribution of total gap junction current, as expressed by AUC, for the 10% of cells with highest or lowest $k_{glc}$ or a random 10% of cells. G). As in E for a bimodal distribution in $k_{glc}$. H). Mean duration of active phase and silent phase averaged over the 10% of cells with highest or lowest oscillation frequency, or a random 10% of cells. I). Mean islet [Ca²⁺] time course showing different portions of the active phase (1–4). J). Mean islet gap junction current during different portions of the active phase, as indicated in I for the 10% of cells with highest or lowest oscillation frequency, or a random 10% of cells. Black lines are fitted regression lines. Error bars are mean ± s.e.m. Repeated measures one-way ANOVA was performed for data in D, E, H to test for significance. Linear regression was performed on data in J. Significance values: ns indicates not significant (p>.05), * indicates significant difference (p < .05), ** indicates significant difference (p < .01), *** indicates significant difference (p < .001), **** indicates significant difference (p < .0001), † indicates significant linear regression (p < .05), ‡ indicated significant linear regression (p < .01). Data representative of 5 simulations with differing random number seeds.

showed a positive (outward, hyperpolarizing) gap junction current, whereas cells with low metabolic activity showed a negative (inward, depolarizing) gap junction current, across all phases of the [Ca²⁺] oscillation (Fig 7E). The magnitude of the gap junction current for less metabolically active cells was also greater. This larger gap junction-mediated current would be expected to hyperpolarize neighboring cells to a greater degree than metabolically active cells depolarizing neighboring cells. Nevertheless, there was significant variability, such that some cells with low metabolic activity had little gap junction current and some cells with high metabolic activity had a positive current that would depolarize neighbors (Fig 7F). When examining the bimodal simulation (Fig 2), we observed broadly similar findings where cells with high metabolic activity depolarize their neighbors whereas cells with low metabolic activity hyperpolarize their neighbors (Fig 7G).

Finally, given the stronger gap junction current associated with the active and silent phases, we analyzed the relationship between the duration of these phases for cells with high and low frequency. Cells with a higher intrinsic oscillation frequency showed both a shorter active phase and shorter silent phase compared to cells with a slower intrinsic oscillation frequency, with there being a greater difference in the active phase (Fig 7H). Interestingly, the whole islet

active and silent phase times were similar to those of cells with a higher oscillation frequency (which on average have lower metabolic activity). During the active phase, the gap junction current was lowest at the beginning of the active phase and greatest just before the downstroke (Fig 7I and 7J). We measured changes to the duration of the active and silent phases after removal of early/late phase cells and low/high frequency cells from Figs 4 and 5. When either late phase cells or low frequency cells were removed, the active phase and duty cycle duration decreased compared to when either early phase cells or high frequency cells were removed, respectively (S10 Fig). Thus, gap junction coupling contributes more to sustaining the active phase compared to initiating the active phase. While slower oscillating cells contribute significantly to setting the islet frequency, given the greater gap junction current, faster oscillating cells may limit the duration of the active phase by terminating the oscillation.

## Discussion

β-cell heterogeneity has largely been studied in single cells. However, recent studies have demonstrated that heterogeneity plays a physiological role in regulating insulin release within the islet [36, 37, 42]. Previously, using computational models and experimental systems, we demonstrated that a large minority (close to 50%) of metabolically active β-cells were necessary to maintain the activity of the islet [42]. In contrast to this, experimental and theoretical studies have suggested that small (~10%) highly functional subpopulations may be required to maintain whole islet [Ca²⁺] dynamics [37, 48]. Here, we investigated the theoretical basis by which small populations of cells may impact islet [Ca²⁺] dynamics.

### Small populations of metabolically active cells are not required to drive elevations in [Ca²⁺]

To determine whether small populations of metabolically active β-cells could drive elevations in [Ca²⁺], we constructed three types of islet simulations: showing either a unimodal normal distribution, skewed distribution or a bimodal distribution in metabolic activity. In each case, we either hyperpolarized the most metabolically active cells or removed them from the simulation. These manipulations are equivalent to those applied in the literature. For example, one study used optical stimulation of eNpHr3.0 to induce a hyperpolarizing Cl⁻ current in 1–10% of cells that showed high levels of [Ca²⁺] coordination and elevated GK [37]. Another study used optical stimulation of ChR2 to induce a depolarizing cation current, with the ~10% of cells activating large parts of the islet showing higher NAD(P)H [36]. In our simulations, we found hyperpolarizing those cells with increased metabolic activity generated similar findings: hyperpolarizing more metabolically active cells silenced the islet to a much greater degree than hyperpolarizing less metabolically active cells. Thus, hyperpolarization or depolarization of metabolically active β-cells can disproportionately suppress or activate islet function, via gap junction coupling.

Importantly, the effects of this targeted silencing were found for both a broad unimodal normal distribution (Fig 1), skewed distribution (Fig 2) and for a bimodal distribution (Fig 2). Within the literature there is not exact consistency in the level of metabolic heterogeneity present. Within dissociated β-cells, a variation of 20–30% in NAD(P)H responses has been observed experimentally [26, 44], and in intact islets a variation of 10–20% has been observed [44]. Instead, ~50% variation is needed to describe experimental observations here. However, early analysis of GK heterogeneity via immunohistochemistry observed substantial variations, which while not quantified would be equivalent to >50% [24]. Similarly, in isolated β-cells the glucose threshold for elevated NAD(P)H varies by ~50% (3-10mM) [26, 49]. This latter study also found a non-normal distribution with ~20% of highly metabolically active β-cells. Thus,

the distributions required in our model to generate results equivalent to experimental observations are broadly feasible. Furthermore, we do note the process of removing β-cells from the islet via dissociation causes cell stress and could disrupt metabolic signatures. Highly metabolically active cells may also be more susceptible to environmental stress [37, 50]. Therefore, further analysis, in situ, is needed to precisely quantify the level of heterogeneity present.

Interestingly, we observed very different results when comparing the effect of targeted hyperpolarization of a set of cells and targeted removal of a set of cells. Hyperpolarizing a small population of metabolically active cells largely silenced the islet, whereas removal of this same cell population had reduced impact. Upon removal, we did observe a moderate reduction in duty cycle of ~40% under a skewed distribution in GK activity, whereas we observed only a small reduction in the duty cycle of ~10% under a bimodal distribution in GK activity. The exact relationship between duty cycle and insulin release in unknown. GK activity is important for setting the Ca²⁺ oscillation frequency and duty cycle, but other downstream elements further modify [Ca²⁺] oscillation dynamics and insulin secretion. For example, pyruvate kinase activation can increase the [Ca²⁺] oscillation frequency and reduce the duty cycle, while amplifying insulin secretion as a result of locally elevating ATP/ADP and closing $K_{ATP}$ channels [51]. Despite the complicated regulation of insulin secretion dynamics, increased [Ca²⁺] duty cycle does correlate with elevated glucose stimulation and insulin release [52, 53]. It has also been suggested that duty cycle and insulin release have a non-linear, sigmoidal relationship [54], thus a ~10% reduction could potentially impact insulin release. Conversely, a ~40% reduction could potentially reduce insulin release to a substantial degree. However, the skewed distribution showed the least correspondence with experimental data when considering hyperpolarizing cell populations, with a smaller difference observed between hyperpolarizing metabolically active cells and inactive cells. As such, the manipulations involving hyperpolarization and cell removal, theoretically, assess the importance a cell has on islet function in different ways. Thus, care must be taken when interpreting the results of optogenetic stimulation-based analysis. Nevertheless, our results imply significant redundancy in the way small populations of cells elevate [Ca²⁺] across the islet (Fig 8).

Cell removal from the simulation may be considered similar to the experimental ablation of that cell. Ablation of small populations of cells that show earlier [Ca²⁺] oscillations, but which overlaps with those cells that show increased [Ca²⁺] coordination, has experimentally been demonstrated to reduce the elevation in [Ca²⁺] across zebrafish islets [38]. These studies showed a substantial reduction in [Ca²⁺] amplitude, whereas our theoretical findings showed no apparent differences in amount of active cells. Little change in [Ca²⁺] activity is observed in the model when removing either those cells with earlier Ca²⁺ oscillations (S5 Fig) or those cells with elevated metabolic activity that when hyperpolarized silences islet [Ca²⁺] (Figs 1 and 2). However, differences do exist between zebrafish islets and mouse islets which our model is based upon and has been validated against, including islet size, gap junction protein isoform and Ca²⁺ dynamics [55, 56]. Thus, species differences may account for these observations.

The way cells interact within our simulated islet is restricted to gap junction electrical coupling. As such, we conclude that gap junction communication is unlikely to be able to explain the role small cell subpopulations play in islet function, under the model assumptions presented here. These conclusions are also consistent with elevated oscillatory [Ca²⁺] being maintained upon a loss of Cx36 gap junction coupling [22], albeit with a lack of synchronization. However, we do note that first phase insulin release is diminished upon a loss of Cx36 gap junction coupling [57]. Therefore, we cannot exclude that small cell subpopulations can drive [Ca²⁺] elevations via gap junction coupling during the initial first phase response.

β-cells can communicate across the islet via paracrine communication. This includes inhibitory factors such as GABA, 5-HT, dopamine and Ucn3 (via δ-cell somatostatin release) and

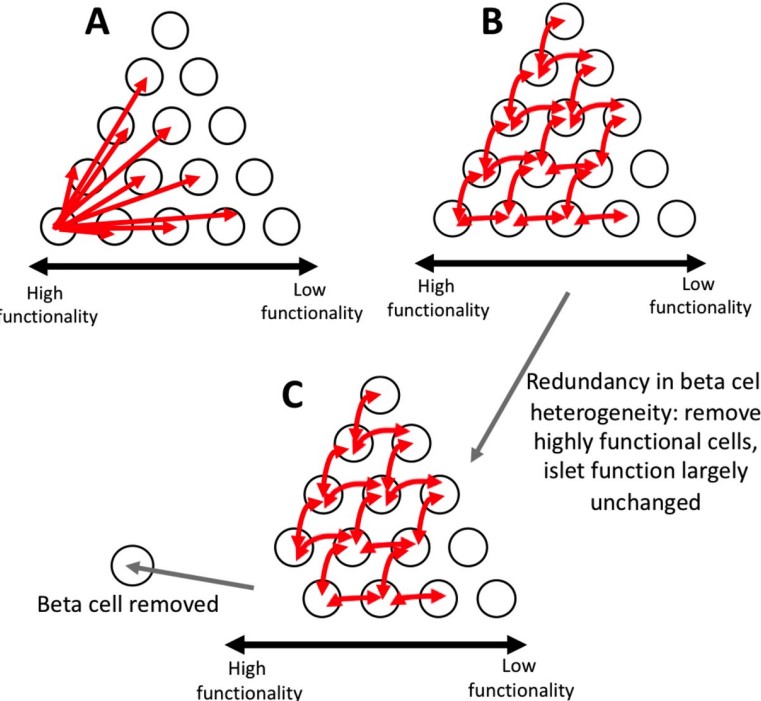

**Fig 8. Schematic of multicellular dynamics of the islet.** A). Schematic of suggestion that small subpopulations of highly functional cells can control whole islet dynamics. White circles represent β-cells. Red arrows represent which cells can be controlled by individual cell where the arrow begins. Increasing functionality in cells is from right to left B). Same as A, but a schematic of how our simulations predict islet [Ca²⁺] dynamics are controlled. Our simulations predict that control is redundant, and many cells can control many other cells. Our simulations predict there is not one small subpopulation that controls the entire islet. C). Same as B, but schematic of how our simulations predict the islet responds when highly functional subpopulations are removed. When highly functional subpopulations are removed, the remaining cells are able to maintain the function of the islet due to the redundancy in control.

stimulatory factors such as ATP [58–60]. Thus, it is conceivable, small subpopulations of metabolic active cells are secreting increased levels of stimulatory paracrine factors. Alternatively, small subpopulations may be acting via other endocrine cells, such as glucagon-secreting α-cells, to stimulate other β-cells within the islet [61]. Removal of immature cell populations can also disrupt islet function, suggesting a broader remodeling of the islet can be induced by small cell subpopulations [62]. Therefore, analyzing whether subpopulations show differential release of paracrine factors will be important to better elucidate their function within the islet.

## Highly metabolic cells have increased connectivity, but this is not due to increased gap junction coupling

Gap junction coupling allows for heterogeneous populations of β-cells to act in a cohesive manner. For example, when populations of normally excitable and inexcitable cells combine within an islet, gap junction coupling ensures that a uniform response occurs, whether this be suppressed [Ca²⁺] or coordinated elevated [Ca²⁺] [41]. Some cell populations have been suggested to have elevated connectivity with other cells in the islet, as measured by correlated [Ca²⁺] oscillations [37, 38], and could result from an increase in gap junction coupling.

In our simulations, highly metabolic cells showed increased connectivity compared with lower metabolically active cells which is in agreement with previous studies showing super connected 'hub' cells have increased GK protein expression. However, when more

metabolically active cells had increased coupling conductance, both highly metabolic cells and low metabolic cells became more similar in their ability to suppress islet function under hyper-polarization (Fig 3). If gap junction coupling is elevated in metabolically active cells, it is reduced in less metabolically active cells. A decrease in coupling lessens how the islet is suppressed in the presence of inexcitable cells that transmit hyperpolarizing current across the islet. Thus, hyperpolarizing a population of metabolically active cells would transmit less hyperpolarizing current beyond the nearest neighbor cells.

We also observed that less metabolically active cells show a greater gap junction current that hyper-polarizes neighboring cells. Thus, there is an asymmetry by which metabolically active and inactive cells within the islet act (Fig 7). As such, increases in coupling are not beneficial for highly metabolic cells to control the islet in a disproportionate manner compared with cells with lower metabolic activity.

Recently, metabolic intermediates have been suggested to diffuse through Cx36 gap junctions in the islet, leading to metabolic coupling [63]. However, this is in disagreement with several studies that Cx36 is strongly selective for cations [64, 65]. Prior modelling studies have also implied that the coordination of slow metabolic oscillations can be described using only electrical coupling [66, 67]. In this study, we did not investigate a role for heterogeneity in regulating slow metabolic oscillations given that highly functional subpopulations have only been characterized in the context of fast electrical dynamics. Further, there is little experimental investigation for the role of electrical coupling in regulating slow [Ca²⁺] and metabolic oscillations. Given that slow metabolic and [Ca²⁺] oscillations likely underlie slow pulsatile insulin release, it would be of interest how functional subpopulations affect metabolic oscillations. Furthermore, if gap junctions within the islet do allow diffusion of metabolic intermediates, this could provide an alternative means by which highly metabolic cells can influence the rest of the islet. Further evidence is needed to test this concept.

## Small subpopulations cannot efficiently act as rhythmic pacemakers

Multiple studies have identified cells that consistently show earlier [Ca²⁺] oscillations that may drive the dynamics of [Ca²⁺] across the islet [36, 38, 47]. These populations have been suggested to have a higher intrinsic oscillation frequency and thus act as a rhythmic pacemaker [36], in the same manner as the cardiac SA node. Here, we investigated whether a small subpopulation of cells with increased oscillation frequency could act as such a pacemaker. We found that cells that show earlier [Ca²⁺] oscillations do have a higher intrinsic oscillation frequency. However, upon removal of these cells, the islet [Ca²⁺] oscillations changed little, suggesting that small populations of these cells are unable to pace islet [Ca²⁺] oscillations. This initially is surprising as with all cells capable of firing, the cell with the highest frequency will depolarize first and stimulate neighbors to fire. However, at least ~30% of high frequency cells are required to even slightly impact islet oscillation frequency. These findings are consistent with prior modelling studies where cells with fast and slow oscillation frequencies, when combined within an islet, led to an overall oscillation midway between the intrinsic cell oscillations [68]. This suggests the oscillation frequency is not per se determined by a small pacemaker population but rather is formed by a weighted combination of all cells across the islet. Thus, the islet also shows significant redundancy where only loss of large populations of cells impacts the activity or dynamics of [Ca²⁺] (Fig 8). Further, the introduction of a small population (~10% cells) with a defined high intrinsic oscillation frequency has little impact on islet [Ca²⁺] oscillations frequency and wave propagation (Fig 6). Heterogeneity in factors other than those considered in our model could also influence [Ca²⁺] oscillation frequency. For example, pyruvate kinase mentioned above locally elevates ATP/ADP and closes K_ATP channels, increasing

the [Ca$^{2+}$] oscillation frequency [51], which is different than the action of increased GK in our model. Further investigation is needed to understand how heterogeneity in other factors could impact the [Ca$^{2+}$] oscillation frequency across the islet.

In contrast to removal of cells that show earlier [Ca$^{2+}$] oscillations, removal of those cells that show delayed [Ca$^{2+}$] oscillations increased the frequency of islet [Ca$^{2+}$] oscillations (Fig 4). These cells on average showed slower oscillations. Therefore, slow [Ca$^{2+}$] oscillations contribute to setting the islet [Ca$^{2+}$] oscillation frequency to a greater degree. Previously, the phantom burster model was shown to have medium bursting modes, under three different conditions, either all fast oscillators, all slow oscillators, or a combination of the two [69]. Our results suggest that slow metabolic oscillations will better coordinate [Ca$^{2+}$] dynamics across the islet, rather than purely a faster-oscillating electrical subsystem. Nevertheless, at least 30% of these slow oscillators are needed to have a substantial impact on the islet dynamics, which is consistent with the oscillation frequency again being formed by a weighted combination of all cells across the islet.

We did not observe a complete overlap between cells that show earlier/delayed [Ca$^{2+}$] oscillations and cells with faster/slower intrinsic [Ca$^{2+}$] oscillations, respectively. Similarly, while removal of the highest and lowest frequency cells changes the overall islet frequency to a greater degree, only removal of cells with delayed [Ca$^{2+}$] oscillations showed a change in frequency above that expected, given the frequency of the remaining cells. Thus, other properties of the islet also contribute to setting the islet oscillation frequency, and these properties remain to be determined.

Therefore, our simulations indicate that there is not a small population of rhythmic pacemaker cells within the islet. Rather, a large number of cells are needed to impact islet frequency. Of interest, the distribution of cells with faster or slower intrinsic [Ca$^{2+}$] oscillations in our simulation is distributed across the islet, whereas cells that show earlier or delayed [Ca$^{2+}$] oscillations exist within a specific region within the islet, often at the islet edge. While having only a minor impact, the spatial distribution of higher frequency cells was important in affecting islet [Ca$^{2+}$] oscillations. Whether intrinsically fast or slow oscillating cells show some spatially restricted distribution is unknown. A different spatial organization could potentially contribute to a greater control over islet frequency, especially if slow oscillators overlap with other properties of the islet that confer greater control over islet oscillation frequency.

We also speculate that the level of gap junction coupling for cells with slower or faster oscillations may be important: the time course of gap junction current indicates that faster oscillating cells transmit a greater hyperpolarizing current to neighboring cells earlier, as compared to slower oscillating cells. This may explain why the islet oscillation active phase duration trends closer to those cells with a higher frequency and thus shorter active phase duration. However, given the lower gap junction current in the silent phase, this appears not to be sufficient to disproportionately impact the oscillation frequency.

## Summary

Overall, the results from this study show how small populations of highly functional cells impact islet function via gap junction electrical coupling. Our simulations suggest that both a small subpopulation of metabolically active cells or the most metabolically active subset of cells within a unimodal distribution are unable to maintain elevated [Ca$^{2+}$] across the islet via gap junction coupling. Further, a small population or subset of cells that shows early [Ca$^{2+}$] elevations or that have a higher oscillation frequency are also unable to act as rhythmic pacemakers to drive oscillatory [Ca$^{2+}$] dynamics. As such the mechanism(s) by which these cells may act to impact islet function should be further investigated.

## Methods

### Coupled β-cell electrical activity model

The coupled β-cell model was described previously [42] and adapted from the published Cha-Noma single cell model [70, 71]. All code was written in C++ and run on the SUMMIT super-computer (University of Colorado Boulder). Example model code is included in supplemental information (S1 Files). All simulations are run at 8mM glucose unless otherwise noted.

The membrane potential ($V_i$) for each β-cell i is related to the sum of individual ion currents as described by [70]:

$$C_m \frac{dv_i}{dt} = I_{Cav} + I_{TRPM} + I_{SOC} + I_{bNSC} + I_{KDr} + I_{KCa(SK)}$$
$$+ I_{K_{ATP}} + I_{NaK} + I_{NaCa} + I_{PMCA} + I_{NaCa} + I_{Coup} \tag{1}$$

Where the gap junction mediated current $I_{Coup}$ [41] is:

$$I_{Coup} = \sum_i g^{ij}_{Coup}(V_i - V_j) \tag{2}$$

Where $g^{ij}_{Coup}$ is the average coupling conductance between cells i and j. Heterogeneity in Cx36 gap junctions is modeled as a γ-distribution with parameters k = θ = 4 as described previously [36] and scaled to an average $g_{Coup}$ between cells = 120pS. The number of cells, N, in each simulation is 1000. The parameters that are heterogenous across all cells in each simulation is described in S1 Table with means and standard deviations.

### Modelling GK activity

The flux of glycolysis $J_{glc}$, which is limited by the rate of GK activity in the β-cell, is described as:

$$J_{glc} = k_{glc} \cdot f_{glc} \cdot ([Re_{tot}] - [Re]) \tag{3}$$

Where $k_{glc}$ is the maximum rate of glycolysis (equivalent to GK activity), which was simulated as a unimodal Gaussian distribution with a mean of 0.000126 ms⁻¹ and standard deviation of 25% of the mean (unless indicated). $[Re_{tot}] = 10mM$, the total amount of pyrimidine nucleotides. The ATP and glucose dependence of glycolysis (GK activity) is:

$$f_{glc} = \frac{1}{1 + \frac{K_{mATP}}{[ATP]_i}} \cdot \frac{1}{1 + \left(\frac{K_G}{[G]}\right)^{hgl}} \tag{4}$$

Where [G] is the extracellular concentration of glucose, hgl is the hill coefficient, $K_G$ is the half maximal concentration of glucose, and $K_{mATP}$ is the half maximal concentration of ATP.

For simulations with changes in variation in GK, the mean remained the same at 0.000126 ms⁻¹, but standard deviation of 1% or 50% of the mean was used.

### Hyperpolarizing cell populations

Hyperpolarization of cells was induced by including a V-independent leak current $I_{hyper}$ that hyperpolarizes the cell [43], described as:

$$I_{hyper} = g_{hyper} * (V - V_{hyper}) \tag{5}$$

Where $g_{hyper}$ is the hyperpolarizing conductance which is zero in the absence of the applied hyperpolarizing current and is $g_{hyper}' (1-p_{0KATP}) \approx g_{hyper}'$ during applied hyperpolarization.

The number of cells that were hyperpolarized were defined as the fraction $P_{hyp}$ multped by the number of cells, N (1000 in all simulations).

### For skewed distribution of GK

A gamma distribution was used to model the unimodal skewed distribution in GK. The gamma distribution has shape parameter k = 1.26 and scale parameter $\theta$ = 0.79. These parameters were fitted to satisfy the following conditions:

$$k_{glc_{GKHigh}} = 3*k_{glc} \tag{6}$$

$$(k_{glc_{GKLow}} * P_{Low} * N + k_{glc_{GKHigh}} * P_{High} * N)/N = k_{glc} \tag{7}$$

Where $k_{glc_{GKHigh}}$ is the mean rate of glycolysis for the GK$^{High}$ population and is 3 times the islet mean, $k_{glc}$ (6). The islet mean $k_{glc}$ remains unchanged from the unimodal normal distribution at 0.000126 ms$^{-1}$ (7). The mean rate of glycolysis for the GK$^{Low}$ population, $k_{glc_{GKLow}}$, is slightly reduced to satisfy this Eq (7). $P_{Low}$, the percent of GK$^{Low}$ cells in the simulation, is 90%, and $P_{High}$, the percent of GK$^{High}$ cells, is 10%. N is the number of cells in the simulation (1000).

### For bimodal distribution of GK

The bimodal distribution of GK was also calculated using Eqs (6) and (7). In this case, there were 2 smaller Gaussian distributions with means $k_{glc_{GKHigh}}$ and $k_{glc_{GKLow}}$ but with standard deviation of $k_{glc_{GKHigh}}$ and $k_{glc_{GKLow}}$ set at 2.5% (10% less than the unimodal normal) of the means to create a distinct bimodal distribution.

### Modelling changes in coupling

Simulations where $k_{glc}$ and $g_{Coup}$ (or $g_{KATP}$) are correlated, the values for $k_{glc}$ and $g_{Coup}$ are calculated using a copula or multivariate normal to calculate correlated value pairs. A correlation of r = 0.7 (or r = -0.7 for inverse correlation of $g_{KATP}$) is used. These values are then transformed to give the variables their appropriate distributions (normal for $k_{glc}$ and gamma distribution for $g_{Coup}$ see Eqs (2–4) for more detail). The paired $k_{glc}$ and $g_{Coup}$ values are then randomly distributed to the cells in the simulation.

For simulations where cells are removed, the conductance, $g_{Coup}$, of the cells to be removed is set to 0 pS. Removed cells are excluded from subsequent islet analysis.

### Determining early and late phase cells

To determine early phase and late phase cells, one full [Ca²⁺] oscillation is taken between time points 300 sec to 400 sec. This time point ensures the model and frequencies are stable and in the second phase of [Ca²⁺] oscillations. A cross correlation is used to determine the time delay of each cell time course compared to the mean [Ca²⁺] across the islet, using xcorr() in MATLAB. A negative delay is therefore equivalent to an earlier oscillation. The early phase cells are determined as the cells with the most negative time delay and the late phase cells are determined as the cells with the most positive time delay. If the cutoff occurs where multiple cells have the same delay, then a random cell is chosen from the cells with the same lag.

### Modelling bimodal distribution for early phase cells

The mean values of the early phase and non-early phase cells in the unimodal normal distribution was used to define a new bimodal distribution as described in S2 Table. All standard

deviations are 1% of the mean. For more information on parameters see [70]. The 'early phase' cell population, $N_{earlyphase}$ comprised 10% of the islet (left column), and the $N_{non-earlyphase}$ comprised the other 90% (right column).

## Network analysis of links

The network analysis was based on previously described methods [72]. The $[Ca^{2+}]$ time course of each cell was correlated with every other cell time course using MATLAB corr() function, to generate a Pearson correlation coefficient matrix for each cell pair. A threshold value of 0.9998 was used to assign a binary value of linked/not linked to each cell pair. The threshold value was chosen to resemble a small world network link distribution, as previously performed [37, 72]. Note, this threshold is higher than that previously used for experimental data (0.75) that generated a small world network link distribution [72]. This is likely due to the differences in precision between experimental fluorescence imaging and a high precision deterministic simulation. Each cell was assigned the total number of links with all other cells and sorted from super-connected (hub) to least connected cells.

## Noise simulations

Stochastic noise was added to the $K_{ATP}$ channel as previously described in which a time varying noise component, S, that follows a normal distribution is added to the $K_{ATP}$ current [43].

$$I_{KATP} = g_{KATP} * [p_{0KATP}(1 + S)] * (V - V_k)$$

Where $p_{0KATP}$ is the open channel probability of the $K_{ATP}$ channel. S is described by

$$S' = \frac{-S}{\tau} - \left(\frac{S}{\tau}\right) + \xi$$

S has a mean of 0 and a standard deviation of ~0.049. $\tau$ = 500ms and $\xi$ is generated from a random number sequence.

## Simulation data analysis

All simulation data analysis was performed using custom MATLAB scripts. The first 1500 time points (150 sec) were excluded to allow the model to reach a stable state.

   Fraction Active was determined by calculating the fraction of cells that were active relative to the total number of simulated cells (1000). Cells were considered active if membrane potential, $[Ca^{2+}]$ exceeded 0.165μM at any point in the time course.

   Duty Cycle was determined as the fraction of the $[Ca^{2+}]$ oscillations spent above a threshold value during the time course analyzed. This threshold value was determined as 50% of the average amplitude of $[Ca^{2+}]$ in an islet simulated at 8mM glucose with 25% variation in GK activity or time above 70% of the maximum $[Ca^{2+}]$ (S10 Fig). Duty cycle was reported as the mean across all cells in the simulated islet.

   Frequency of a cell in the islet was determined by taking the $[Ca^{2+}]$ time-course between times 150 sec and 400 sec and identifying the first 2 peaks. The peak-to-peak time was determined, and this oscillation period was inverted to calculate the frequency. For whole islet frequency calculations, the coupling in the islet is $g_{Coup}$ = 120pS and the mean islet frequency is calculated over all cells in the simulation.

   Intrinsic frequencies were determined using simulations where the mean coupling conductance of all cells in the simulation is $g_{Coup}$ = 0pS so that all cells oscillate on their own without

influence from other cells within the simulation. When determining low and high frequency cells in the simulation, only active cells were used.

<u>Expected Frequency</u> was determined by finding the average intrinsic frequencies of the cells ($g_{Coup}$ = 0pS) that are included in the simulation. These values are then compared to the simulation where $g_{Coup}$ = 120pS.

<u>Total gap junction current</u> for a cell was calculated by summing the gap junction current over each connection between the cell and all of its neighbors, as in Eq (2). The total membrane current was calculated as the sum over each current for that cell, as in Eq (1).

<u>Active, silent, upstroke and downstroke phases</u> were chosen manually. The Area Under the Curve (AUC) was calculated using the trapz() function in MATLAB, which calculates trapezoidal integration over the time period. AUC was calculated for each cell in the given decile and then averaged over those cells.

<u>Active phase duration</u> for one oscillation was determined for each cell, as the total time [Ca²⁺] was above 70% of the maximum value, divided by the number of oscillations over the duration assessed. The silent phase duration was similarly calculated as the total of time [Ca²⁺] was below 40% of maximum value.

## Statistical analysis

All statistical analysis was performed in Prism (GraphPad). Either a Student's t-test (or Welch's t-test for significantly difference variances) or a one-way ANOVA with Tukey post-hoc analysis was utilized to test for significant differences for simulation results. Paired t-test or repeated measures ANOVA was used anywhere the results were compared with a simulated matching control islet or groups within the same islet, e.g., before a population was either hyperpolarized or uncoupled. Data is reported as mean ± s.e.m. (standard error in the mean) unless otherwise indicated.

## Supporting information

**S1 Fig. Histograms of GK activity ($k_{glc}$) and $g_{Coup}$ for all unimodal normal, unimodal skewed and bimodal distributions in GK activity for Figs 1–3.** A). All unimodal normal distributions' histograms. Left: Average frequency of cells at varying GK rate ($k_{glc}$) for simulations that have different standard deviations in GK activity from Fig 1. Right: Corresponding histogram of average frequency of cells at varying coupling conductance ($g_{Coup}$). B). As in A but for simulations with a skewed normal distribution of GK activity from Fig 2A–2E. C). As in A but for simulations with a skewed normal distribution of GK activity and correlated GK and $g_{Coup}$ activity from Fig 3A–3C. D). As in A but for simulations with a bimodal distribution of GK activity from Fig 2I–2K. E). As in A for simulations with bimodal distribution of GK activity and correlated GK and $g_{Coup}$ from Fig 3D–3F. Data representative of 5 simulations with differing random number seeds.
(TIF)

**S2 Fig. Effects of noise on hyperpolarization-induced cell silencing.** A). Fraction of cells showing elevated [Ca²⁺] activity (active cells) vs. the percentage of cells hyperpolarized in islet from simulations with a unimodal normal distribution with 25% variation in GK activity ($k_{glc}$). Simulations run in the presence of stochastic noise (see methods). B). As in A but for simulations with 50% variation in GK activity. C). As in A but for simulations with unimodal skewed distribution. Error bars are mean ± s.e.m. Repeated measures one-way ANOVA with Tukey post-hoc analysis was performed for A and B. Student's paired t-test was performed to test for significance in C. Significance values: ns indicates not significant (p>.05), * indicates

significant difference (p < .05), ** indicates significant difference (p < .01), *** indicates significant difference (p < .001), **** indicates significant difference. Data representative of 5 simulations with differing random number seeds.
(TIF)

**S3 Fig. Additional simulations with unimodal normal distribution in GK activity with correlated g$_{Coup}$ and g$_{KATP}$.** A). Scatterplot of g$_{Coup}$ vs. k$_{glc}$ for each cell from a representative simulation where g$_{Coup}$ is correlated with k$_{glc}$ for simulation where GK activity is a modeled as a unimodal normal distribution. B). Fraction of cells showing elevated [Ca²⁺] activity (active cells) vs. the percentage of cells hyperpolarized in islet from simulations with a unimodal normal distribution in k$_{glc}$ with correlated g$_{Coup}$ and k$_{glc}$ as in A. Hyperpolarized cells are chosen based on their GK rate which is correlated to g$_{Coup}$. C). As in B. but comparing hyperpolarization in high GK cells in the presence (B) and absence ([Fig 1C]) of correlations in g$_{Coup}$. D). as in A but from a simulation where g$_{Coup}$ and k$_{glc}$ and g$_{KATP}$ (K$_{ATP}$ channel conductance) are correlated. E). As in B. but for simulations where g$_{Coup}$ and k$_{glc}$ and g$_{KATP}$ are correlated. F). As in C. but comparing high GK cells hyperpolarization from [Fig 1C] to high GK hyperpolarization from simulations where g$_{Coup}$ and k$_{glc}$ and g$_{KATP}$ are correlated (E). Error bars are mean ± s.e.m. Repeated measures one-way ANOVA with Tukey post-hoc analysis was performed for simulations in B and C (if there were any missing values a mixed effects model was used) and a Student's t-test was performed for C and F (Welches t-test for unequal variances was used when variances were determined to be statistically different using an F-test) to test for significance. Significance values: ns indicates not significant (p>.05), * indicates significant difference (p < .05), ** indicates significant difference (p < .01), *** indicates significant difference (p < .001), **** indicates significant difference (p < .0001). Data representative of 5 simulations with differing random number seeds.
(TIF)

**S4 Fig. Simulations predicting effect of 50% reduction in coupling in simulations with unimodal normal and bimodal distributions in GK activity.** A). Fraction of cells showing elevated [Ca²⁺] activity (active cells) vs. the percentage of cells hyperpolarized in islet from simulations with a unimodal normal distribution as in [Fig 1C] but with 50% reduction in average coupling conductance (60pS) for all cells. Hyperpolarized cells are chosen based on their GK rate. B). As in A. but comparing hyperpolarization in high GK cells in simulations with full coupling (120pS–[Fig 1C]) and reduced coupling (60pS) from A. C). as in A but for bimodal simulations with reduced coupling (60pS). D). As in B but comparing bimodal distributions in GK with full coupling (120pS) from [Fig 2J] to bimodal simulations with reduced coupling (60pS) from C. Error bars are mean ± s.e.m. Student's paired t-test was performed to test for significance for all simulations. Significance values: ns indicates not significant (p>.05), * indicates significant difference (p < .05), ** indicates significant difference (p < .01), *** indicates significant difference (p < .001), **** indicates significant difference. Data representative of 4–5 simulations with differing random number seeds.
(TIF)

**S5 Fig. Fraction of active cells in simulations where cells are uncoupled from the rest of the cells in the islet from Figs [4]–[6].** A). Fraction of cells showing elevated [Ca²⁺] activity (active cells) in simulated islets vs. the percentage of cells uncoupled in islet from simulations in [Fig 4]. B). As in A but for simulations in [Fig 5]. C). As in A but for simulations in [Fig 6]. D). As in A but for simulations in [S8 Fig]. Error bars are mean ± s.e.m. Repeated measures one-way ANOVA was performed for simulations in A and B and a Student's paired t-test was performed for C and D to test for significance. Significance values: ns indicates not significant

(p>.05), * indicates significant difference (p < .05), ** indicates significant difference (p < .01), *** indicates significant difference (p < .001), **** indicates significant difference. Data representative of 5 simulations with differing random number seeds.
(TIF)

**S6 Fig. Random removal of cells vs. random removal of a region of cells.** A). Schematic showing which cells are chosen to be removed when a random selection of cells is chosen across the islet. B). Schematic showing which cells are chosen to be removed when a *random region* of cells is chosen. C). The frequency of the islet after removal of 0%, 10%, or 30% of randomly chosen cells or from a random region. Error bars are mean ± s.e.m. Student's t-test was performed for 10% and a Welch's t-test for unequal variances was used to test for significance at 30% of cells removed. Significance values: ns indicates not significant (p>.05), * indicates significant difference (p < .05), ** indicates significant difference (p < .01), *** indicates significant difference (p < .001), **** indicates significant difference. Data representative of 4–9 simulations with differing random number seeds.
(TIF)

**S7 Fig. Simulations predicting the effect of 50% reduction in coupling in simulations where early and late phase cells are removed under a unimodal normal model.** A). Average frequency of islet when indicated populations of cells are removed from the simulated islet with 50% reduction in coupling conductance (60pS). B). Change in frequency of islet with indicated populations removed with respect to control islet with all cells present. C). Change in frequency when early phase cells are removed compared to average oscillation frequency of remaining cells that indicates the expected oscillation frequency. D). Same as C. but for simulations where late phase cells are removed. Error bars are mean ± s.e.m. Repeated measures one-way ANOVA with Tukey post-hoc analysis was performed for simulations in A-B and a Student's paired t-test was performed for C and D to test for significance. Significance values: ns indicates not significant (p>.05), * indicates significant difference (p < .05), ** indicates significant difference (p < .01), *** indicates significant difference (p < .001), **** indicates significant difference (p < .0001). Data representative of 4 simulations with differing random number seeds.
(TIF)

**S8 Fig. Simulations predicting the effect of removing cells from individual populations of the bimodal model of early phase cells.** A). Average intrinsic oscillation frequencies of all cells, top 1% or 10% of high frequency cells, or low frequency cells when re-simulated in the absence of gap junction coupling from bimodal model of phase. B). Phase lag from islet average of top 1% or 10% of high frequency, low frequency cells, or random cells. C). Average $k_{glc}$ from all cells, high frequency cells, or low frequency cells across simulated islet. D). Change in frequency of islet with indicated populations removed with respect to control islet with all cells present. E). Change in frequency when high frequency cells are removed compared to average oscillation frequency of remaining cells that indicates the expected oscillation frequency. F). Same as E. but for simulations where low frequency cells are removed. Error bars are mean ± s.e.m. Repeated measures one-way ANOVA with Tukey post-hoc analysis was performed for simulations in A-D and a Student's paired t-test was performed for E and F to test for significance. Significance values: ns indicates not significant (p>.05), * indicates significant difference (p < .05), ** indicates significant difference (p < .01), *** indicates significant difference (p < .001), **** indicates significant difference (p < .0001). Data representative of 5 simulations with differing random number seeds.
(TIF)

**S9 Fig. Simulations predicting the effect of removing a region of high frequency cells from a bimodal model of early phase cells.** A). Schematic of frequency across simulated islet with a bimodal distribution in GK activity and a region of high frequency cells. B). Schematic of phase lag across simulated islet with a bimodal distribution in GK activity and a region of high frequency cells. C). Change in frequency of islet with indicated populations removed with respect to control islet with all cells present comparing bimodal model with a region of high frequency cells to a bimodal model with randomly distributed high frequency cells as in Fig 6. D). Change in frequency when high frequency region is removed compared to average oscillation frequency of remaining cells that indicates the expected oscillation frequency. Error bars represent mean ± s.e.m. Student's t-test was performed for C and D (paired test) to test for significance. Significance values: ns indicates not significant (p>.05), * indicates significant difference (p < .05), ** indicates significant difference (p < .01), *** indicates significant difference (p < .001), **** indicates significant difference (p < .0001). Data representative of 5 simulations with differing random number seeds.
(TIF)

**S10 Fig. Analysis of changes in [Ca2+] wave dynamics when early/late phase or high/low frequency cells are removed from islet.** A). Change in mean duration of active phase when top 1%, 10% or 30% early/late phase cells are removed from simulations in Fig 4. B). Change in mean duration of silent phase when top 1%, 10% or 30% early/late phase cells are removed from simulations in Fig 4. C). Change in mean duty cycle when top 1%, 10% or 30% early/late phase cells are removed from simulations in 4. D). As in A for simulations when high/low frequency cells are removed from Fig 5. E). As in B for simulations when high/low frequency cells are removed from Fig 5. F). As in C for simulations when high/low frequency cells are removed from Fig 5. Error bars are mean ± s.e.m. Paired Student's t-test was used to test for significance. Significance values: ns indicates not significant (p>.05), * indicates significant difference (p < .05), ** indicates significant difference (p < .01), *** indicates significant difference (p < .001), **** indicates significant difference (p < .0001). Data representative of 5 simulations with differing random number seeds.
(TIF)

**S1 Table. Heterogenous parameters in computational islet model.** Table describes the parameters in the computational model that are heterogenous for each cell. The mean and standard deviation is defined in the table. Changes to these parameter distributions are discussed in the methods.
(PDF)

**S2 Table. Parameters for bimodal early phase cell simulations.** Table describes the parameters that have heterogeneous populations in computational model. The mean of each population is determined from the mean parameter value from unimodal normal simulations (See methods).
(PDF)

**S1 Files. Model code used in this study, in zip file.** Files include those used to generate data in Figs 1, 2, 4 and 6.
(ZIP)

## Acknowledgments

The authors thank Dr David J Hodson (University of Birmingham, UK) and Dr Victoria Salem (Imperial College London, UK) for reviewing this manuscript and for providing helpful

comments and suggestions. The authors are also grateful for utilization of the SUMMIT super-computer from the University of Colorado Boulder Research Computing Group, which is supported by the National Science Foundation (awards ACI-1532235 and ACI-1532236), the University of Colorado Boulder, and Colorado State University.

## Author Contributions

**Conceptualization:** JaeAnn M. Dwulet, Richard K. P. Benninger.

**Data curation:** JaeAnn M. Dwulet, Richard K. P. Benninger.

**Formal analysis:** JaeAnn M. Dwulet, Jennifer K. Briggs.

**Funding acquisition:** JaeAnn M. Dwulet, Richard K. P. Benninger.

**Investigation:** JaeAnn M. Dwulet, Jennifer K. Briggs.

**Methodology:** JaeAnn M. Dwulet, Jennifer K. Briggs, Richard K. P. Benninger.

**Project administration:** Richard K. P. Benninger.

**Resources:** JaeAnn M. Dwulet.

**Software:** JaeAnn M. Dwulet.

**Validation:** JaeAnn M. Dwulet.

**Visualization:** JaeAnn M. Dwulet.

**Writing – original draft:** JaeAnn M. Dwulet, Richard K. P. Benninger.

**Writing – review & editing:** JaeAnn M. Dwulet, Richard K. P. Benninger.

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
