## [Decision Letter · Decision Letter 0]

28 Oct 2020

Dear Assoc. Prof. Benninger,

Thank you very much for submitting your manuscript "Small subpopulations of β-cells do not drive islet oscillatory [Ca2+] dynamics via gap junction communication" for consideration at PLOS Computational Biology.

As with all papers reviewed by the journal, your manuscript was reviewed by members of the editorial board and by several independent reviewers. In light of the reviews (below this email), we would like to invite the resubmission of a significantly-revised version that takes into account the reviewers' comments.  Note that reviewers 2 and 3 raised methodological concerns that will be critical to address.  Also, the reviewers are split about the breadth of the audience that this paper will find, so you are encouraged to consider whether additions can be made (still within the scope of the project) that would broaden its appeal, and to document any such updates.

We cannot make any decision about publication until we have seen the revised manuscript and your response to the reviewers' comments. Your revised manuscript is also likely to be sent to reviewers for further evaluation.

Sincerely,

Jonathan Rubin

Associate Editor

PLOS Computational Biology

Jason Haugh

Deputy Editor

PLOS Computational Biology

Reviewer's Responses to Questions

**Comments to the Authors:**

Reviewer #1: This is a very important and timely paper that appears carefully executed, I am favorably impressed. The paper uses a network model of the pancreatic islet to argue, in a quantitative way, that islet subpopulations cannot exert strong control over calcium dynamics via gap junctions. This issue was raised by Rorsman, Satin, and Rutter in two recent and highly charged Perspectives in Diabetes, and this paper is exciting because it provides a quantitative platform for (and against) their arguments. The model seems generalizable and robust, accommodating a range of biological possibilities with the same general conclusions.

Major

1. The authors show that hyperpolarizing 10% of cells with high GK flux suppressed islet calcium, whereas hyperpolarizing cells with normal metabolic activity had no effect. This can only occur if there is a physiological relationship between high glucose oxidation and higher coupling to neighbors, particularly as removing these highly metabolic cells altogether did not suppress calcium. It may not be the metabolism of these cells that matters, per se, but the coupling. What is the experimental justification for linking high GK activity (directly or inversely) to coupling conductance? Is there any? Or is this an emergent property of the network?

2. Related to Results, Lines 264-266: “These cells [cells that fire before the rest of the islet] have been suggested to have higher intrinsic oscillation frequency [29, 40], which may lend themselves to act as rhythmic pacemakers to drive [Ca2+] oscillations across the islet.”

a) A minor point is that Reference 40 does not appear to addresses the relationship between frequency and pacemaking.

b) It seems wholly speculative that first responder cells would, on their own, possess higher frequency (as suggested in Line 274). If there is experimental evidence for this, please provide it.

c) Most importantly, why would high frequency cells lend themselves to pacemaking when increasing GK activity lowers oscillation frequency (as stated in Lines 272-274)?

Minor

1. As written, the short title isn’t accurate and should be modified – even single cells can generated calcium oscillations.

2. While Lines 90-101 of the introduction do a nice job of introducing the field, I’m not thrilled with the abstract. Don’t get me wrong, I greatly appreciate the neutral tone taken by the authors, however they have divorced their findings from the experiments that motivate this paper, and therefore undersell the papers significance:

a) There is an important experimental rationale for these modeling studies that is told in introduction but not the abstract. The authors did not choose these specific manipulations of subpopulations at random – can the backstory that motivates this paper be somehow acknowledged in the abstract to enhance the impact?

b) The authors run simulations with both removal and hyperpolarization of highly metabolic cells. It could be clarified that hyperpolarization mimic optogenetic silencing. What experiment does removal mimic, laser ablation?

c) Abstract, lines 29-31: It is not clear what is meant by “increased excitability”. It is also not clear whether the authors refer to slow metabolic or fast electrical oscillations, which interact and both contribute to excitability.

d) Abstract, line 32: “metabolic activity” should be referred to more specifically as glucose oxidation (assuming the authors mean changes in GK flux).

3. Introduction, line 94: 10% of beta cells are highly excitable… and had higher metabolic activity [29]. Excitable at lower glucose (recruitable?) Higher NADH response to glucose?

4. Results, Lines 112-121: The first paragraph of the Results in largely redundant with the introduction.

5. Lines 76-77: References 9-11 do not do justice to the statement that “disruptions to this GSIS pathway occur in diabetes”.

6. Line 508: “Duty cycle is a large determinant of insulin release, thus a ~10% reduction would not be expected to impact insulin release substantially.” This statement is misleading and almost certainly false.

7. Line 593: “This also suggests that slow metabolic oscillations will better coordinate [Ca2+] dynamics across the islet, rather than purely a faster-oscillating electrical subsystem.” Haven’t prior modeling studies by Sherman and colleagues already shown this? If so they should be cited in the Discussion.

Reviewer #2: Review of: Small subpopulations of β-cells do not drive islet oscillatory [Ca2+] dynamics via gap junction communication, by Dwulet, Briggs and Benninger.

Summary

The authors present a computational study investigating the ability of subpopulations of beta cells in an islet to control multicellular islet dynamics. A previously validated computational multicellular model of the islet is used, with a focus on different patterns of heterogeneity in two key parameters: GK activity (kglc) and gap-junctional coupling strength (gcoup). Islets with combinations of unimodal and bimodal distributions for these parameters were generated, as were situations with correlations between parameters. The islet activity response to the hyperpolarization or removal/decoupling of select subsets of cells was then examined. The the possibility of pacemaker subpopulations was also examined by modeling the effect of decoupling subsets of cells with high or low frequency (low-phase/high-phase). Finally, the gap junction currents in a heterogeneous islet were examined.

With a unimodal distribution of GK and gcoup, selectively hyperpolarizing the GK-high subset was more effective at reducing the fraction of active islet cells than the GK-low, requiring removal of fewer cells. Decoupling subsets of either GK-high or GK-low had little effect on active fraction of remaining islet cells.

With a bimodal distribution of GK, removal of GK-high cells required <=10% of islet cells to silence the islet, more effective than the unimodal case. However, the remaining cells had a reduced mean GK by construction, to keep the islet-mean GK constant. Decoupling of the 10% GK-high cells did not abolish activity in the remaining cells, but the islet had reduced duty cycle (teh mean GK of remaining cells is reduced).

Imposing correlations between GK and gcoup reduced the islet-silencing efficacy of hyperpolarization, due to the weakend coupling in remaining cells. Again, by construction the low-gcoup cells in the bimodal gcoup case had reduced gcoup compared to the unimodal gcoup case to keep the mean constant, as was done for GK.

Phase analysis was used to identify fast/slow cells and subpopulations of these were removed. The effects of these perturbations were small, essentially because the remaining cells retained oscillatory capability and were sufficiently coupled. Finally, the analysis of gap-junctional currents showed that high-frequency and low-frequency cell subpopulations had gap-junction currents that acted to bring these cells closer to the mean activity.

From these results the authors conclude that, in the context of the modeling choices made: 1) small subpopulations of metabolically active cells are not required for islet activity, 2) gap junctional coupling reduces the impact of specialized subpopulations, and 3) subpopulations cannot efficiently act as pacemakers for the islet.

Recommendation

For complex systems such as pancreatic islets, computational simulations such as those presented here are crucial tools for testing and generating hypotheses. The focus of this manuscript, whether specialized subpopulations can control islet activity, is important and timely given recent developments and significant ongoing debate in the field. However, I have some concerns relating to methods used to generate parameter sets for the computational islets in the study. The cases of parameter bimodality and correlation seem unrealistic, and some of the main results seem to reflect artefacts in how these parameter distributions were generated. The conclusions drawn by the authors are supported by the results presented, but they are tightly tied to the parameter choices. I believe these parameter distributions could be improved in a straightforward way, but this would involve potentially significant changes to the results and possibly conclusions of the paper. Therefore, in its current form I cannot recommend the manuscript for publication.

Major comments

The method for generating correlated parameter distributions gives a highly artificial distribution of parameter combinations, which is the direct cause of the results observed in those simulations. A more realistic method will give 2D random variables, which as a scatter-plot will be a cloud of points in Kglc-gcoup parameter space with a specified correlation. This is easily achieved given the two marginal distributions for GK (Normal) and gcoup (gamma) using a copula. See how to do this in Matlab here: https://www.mathworks.com/help/stats/copulas-generate-correlated-samples.html

I also attached an example figure and Matlab code showing the result I'm imagining instead of the values shown in FigS2A. A similar approach should be used for other simultions with correlated parameter distributions (e.g., Fig3, FigS2B). Note that the same code easily generates negatively correlated (rho<0) or uncorrelated (rho=0) distributions. I suspect this will give more interesting results - there will be a more natural distribution of GK and gcoup parameter values across the islet.

Another concern regarding the selected parameter distributions is the sharp, highly separated bimodal distributions: these seem unrealistic. Furthermore, the constraint of keeping the global distribution's mean constant seems to introduce undesired artefacts. For example, in the bimodal kglc case, lowering the mean kglc of GK-low cells, by construction, makes the islet more susceptible to silencing than the default (Fig1) case, because 90% of cells have lower kglc. Similarly, reducing the mean gcoup in the low-gcoup group, the islet is less coupled overall and thus less sensitive to interventions done on the GK-high/high-gcoup group. Is such a sharp bimodal distribution desired and/or realistic? An alternative is to compare a right-skewed distribution for Kglc to the Normal distribution case (perhaps gamma, or some distribution motivated by data on GK activity levels if available)

The parameter sets shown in the figure seem to have very tight distributions. By comparison to Fig1E/FigS1A, the distributions for Kglc in Figures Fig2B, Fig3A/D, and FigS1B-D all appear to show standard deviation of 1% of the mean, but the text claims simulations were done with 25%. The exception is Fig2A, which appears as the stated 25% deviation for Kglc.

Other comments

Hyperpolarization vs decoupling are different conditions, which could be made clear early in the manuscript instead of just in the discussion. Hyperpolarization of a cell (or cell subpopulation) will generate inhibitory currents in neighbor cells via gap junctions, while decoupling does not. This could be used as the guiding principle for explaining the results of the paper.

Terminology suggestion for parameter distributions: "unimodal" instead of "continuous" for parameter distributions.

Terminology suggestion for phase analysis: early/late phase.

For all bar-charts, the data points per simulation could also be shown to give an idea of the variability (as was done in FigS5)

From the simulation code, it appears that variation in several cell parameters was also included (e.g, gKATP). This is also implied in the description of modeling low/high phase cells. It would be helpful to list these parameters with heterogeneity clearly in the first section of Methods, even if they are the same as prior modeling studies with the same model.

The text does not clearly indicate which value is used for computation of Icoup in a given cell - the value of g_coup^(i,j) isn't defined. In the code for the simulations, it is clear that the average value of gcoup from cell i and j is used to compute Icoup. This was described in previous manuscripts using this model (eg. Westacott et al 2017), and it would be helpful describe it in the methods section here as well.

The average gcoup is claimed to be ~120pS in the text, but Fig3A/D, FigS1, and FigS2A shows ~0.12pS. Should the yaxis show nS instead of pS?

Why is ghyper = ghyper'*(1-p0_KATP) when hyperpolarization is on? Shouldn't this be an ohmic current independent of KATP channel open probability, for example as a simple model of a photo-activated chloride current?

It seems that the low/early-phase cells clustered in the same region probably because the simulation is deterministic. If noise is introduced (e.g., randomized initial conditions; a noise current in the voltage equation for each cell) does the location of low-phase change (possibly from burst to burst)? If there are sufficiently many low-phase cells with very similar intrinsic frequencies, it seems this could be possible, but if there is a clearly dominant "lowest-phase" region, even noise may not be able to shift the burst-initiation site.

The coupling currents are defined by the voltage difference between cells, so whether the current is inward or outward is defined by which cell is leading vs lagging - gap juntions will try to bring the two voltages towards each other. Perhaps a more informative analysis than characterization of whether subpopulations have inward/outward currents during different phases of a burst would be to consider the value of gcoup relative to the cell's total conductance. A more significant effect of coupling will occur when the gouping conductance is relatively large compared to the cell's total conductance at that moment. This would be particularly important for cells at rest with low total conductance in that state.

Reviewer #3: The review is uploaded as an attachment.

**Have all data underlying the figures and results presented in the manuscript been provided?**

Reviewer #1: Yes

Reviewer #2: Yes

Reviewer #3: Yes

PLOS authors have the option to publish the peer review history of their article (what does this mean?). If published, this will include your full peer review and any attached files.

Reviewer #1: No

Reviewer #2: No

Reviewer #3: No
---

## [Decision Letter · Decision Letter 1]

18 Mar 2021

Dear Assoc. Prof. Benninger,

Thank you very much for submitting your manuscript "Small subpopulations of β-cells do not drive islet-wide oscillatory [Ca2+]  dynamics via gap junction communication" for consideration at PLOS Computational Biology. As with all papers reviewed by the journal, your manuscript was reviewed by members of the editorial board and by several independent reviewers. The reviewers appreciated the attention to an important topic. Based on the reviews, we are likely to accept this manuscript for publication, providing that you modify the manuscript according to the review recommendations.

Note that these recommendations are quite minor and some are left to your discretion.  It is unlikely that the revised manuscript will go back to the reviewers; if you simply address these comments and accompany your resubmission with a brief response letter summarizing your updates, then we should be able to render a rapid new decision.

Sincerely,

Jonathan Rubin

Associate Editor

PLOS Computational Biology

Jason Haugh

Deputy Editor

PLOS Computational Biology

[LINK]

Reviewer's Responses to Questions

**Comments to the Authors:**

Reviewer #1: I have only two minor comments:

1) Introduction, line 74: comma not semicolon.

2) Discussion, line 550: “Increased [Ca2+] duty cycle correlates with elevated glucose stimulation and insulin release [51, 52], thus a ~10% reduction would not be expected to impact insulin release substantially.”

There is evidence to the contrary. A 10% change in duty cycle may well be meaningful for insulin secretion (PMID 7703919). Furthermore, the relationship between GK, duty cycle, and exocytosis is more complicated than the authors present. In their discussion the authors might should acknowledge that duty cycle and oscillation period, while initially set by GK, are then modified downstream by PK, which reduces duty cycle and period while at the same time increasing the steepness of the correlation between GK activity and exocytosis (PMID 33147484).

This comment also applies to Lines 664/665 that discuss oscillation frequency.

Reviewer #2: The revisions done by the authors have significantly improved the manuscript, and have clarified the conclusions of the study. The modeling has been done carefully, and represents well the current experimental knowledge of the relevant beta cell and islet properties. It is noteworthy that despite the changes (improvements) to the parameter distributions used in simulations, the main conclusions of the manuscript remain unchanged. This indicates that the results are robust to modeling choices, and strengthens the conclusions. All my major concerns have been addressed.

I have only a few more comments/suggestions that are largely cosmetic, which I leave to the authors to adopt or not based on their preference.

Line 106 - By "highly excitable" do you intend to refer to "highly metabolically active"? In beta cells, higher metabolic activity should lead to higher excitability (via KATP channels), but there could be non-metabolic reasons for higher/lower excitability. In any case, it seems useful here to specifically mention "highly metabolically active cells" as they (high GK cells) are a main focus of the manuscript.

Line 104 + 107 - Two sentences begin with "This includes"; one of them could be reworded. E.g., 107 could start with something like "We systematically examined the effects of removal of ..."

End of introduction - A sentence or two at the end of the final paragraph indicating the major take-home message would fit nicely. This could periphrase the final portion of the Abstract.

Fig 4D, 5F, 6F - Is there a reason why kglc needs to be normalized? If so indicate this in the legends, or at least indicate that kglc is normalized in the legends.

Line 655; 665 - "Our results suggest" instead of "suggests"; "properties remain" instead of "remains"

Reviewer #3: The revised manuscript addresses the previuos concerns so that it can be accepted in the current form.

**Have all data underlying the figures and results presented in the manuscript been provided?**

Reviewer #1: Yes

Reviewer #2: Yes

Reviewer #3: None

PLOS authors have the option to publish the peer review history of their article (what does this mean?). If published, this will include your full peer review and any attached files.

Reviewer #1: No

Reviewer #2: No

Reviewer #3: **Yes: **Simonetta Filippi

Figure Files:

Data Requirements:

Reproducibility:

References:

---

## [Editor Report · Decision Letter 2]

7 Apr 2021

Dear Assoc. Prof. Benninger,

We are pleased to inform you that your manuscript 'Small subpopulations of β-cells do not drive islet-wide oscillatory [Ca2+]  dynamics via gap junction communication' has been provisionally accepted for publication in PLOS Computational Biology.

Best regards,

Jonathan Rubin

Associate Editor

PLOS Computational Biology

Jason Haugh

Deputy Editor

PLOS Computational Biology

---

## [Editor Report · Acceptance letter]

27 Apr 2021

PCOMPBIOL-D-20-01679R2 

Small subpopulations of β-cells do not drive islet-wide oscillatory [Ca2+]  dynamics via gap junction communication

Dear Dr Benninger,

I am pleased to inform you that your manuscript has been formally accepted for publication in PLOS Computational Biology. Your manuscript is now with our production department and you will be notified of the publication date in due course.

With kind regards,

Katalin Szabo
